# Targeting TGFβR2-mutant tumors exposes vulnerabilities to stromal TGFβ blockade in pancreatic cancer

Huocong Huang[1] iD, Yuqing Zhang[1,2] iD, Valerie Gallegos[3], Noah Sorrelle[1], Mohamed Medhat Zaid[4], Jason Toombs[1], Wenting Du[1,2], Steven Wright[1], Moriah Hagopian[5], Zhaoning Wang[6], Abdel Nasser Hosein[7], Adwait Amod Sathe[8], Chao Xing[8], Eugene J Koay[4], Kyla E Driscoll[9] & Rolf A Brekken[1,2,5,*] iD

## Abstract

TGFβ is important during pancreatic ductal adenocarcinoma (PDA) progression. Canonical TGFβ signaling suppresses epithelial pancreatic cancer cell proliferation; as a result, inhibiting TGFβ has not been successful in PDA. In contrast, we demonstrate that inhibition of stromal TGFβR2 reduces IL-6 production from cancer-associated fibroblasts, resulting in a reduction of STAT3 activation in tumor cells and reversion of the immunosuppressive landscape. Up to 7% of human PDA have tumor cell-specific deficiency in canonical TGFβ signaling via loss of TGFβR2. We demonstrate that in PDA that harbors epithelial loss of TGFβR2, inhibition of TGFβ signaling is selective for stromal cells and results in a therapeutic benefit. Our study highlights the potential benefit of TGFβ blockade in PDA and the importance of stratifying PDA patients who might benefit from such therapy.

**Keywords** cancer-associated fibroblast; IL-6; pancreatic cancer; TGFβ; tumor immunology

**Subject Categories** Cancer; Digestive System

See also: **RM Carr & ME Fernandez-Zapico** (November 2019)

## Introduction

Pancreatic ductal adenocarcinoma (PDA) is a lethal cancer with a 5-year survival rate of ~8%, and it is projected to become the second leading cause of cancer-related deaths in the United States by the year 2030 (Rahib *et al*, 2014). A substantial feature of PDA is a dense desmoplastic reaction that leads to an accumulation of fibrotic tissue (Feig *et al*, 2012). The desmoplastic landscape consists of heterogeneous stromal cell populations that include fibroblasts, macrophages, endothelial cells and lymphocytes. This complex tumor microenvironment contributes to PDA development, invasion, metastasis, immune evasion, and resistance to therapies.

The transforming growth factor beta (TGFβ) signaling pathway is an evolutionarily conserved regulator of a wide variety of biological processes (Massague, 2012; David & Massague, 2018), such as embryonic development, wound healing, and inflammation. Canonical TGFβ signaling is driven by the activation of the TGFβ receptor system (TGFβR1/R2), which results in phosphorylation of the receptor-regulated SMAD proteins (SMAD2 and SMAD3) that stimulate transcription of target genes in association with SMAD4. In PDA, TGFβ signaling has multifaceted and cell type-specific effects that contribute to many aspects of the tumor microenvironment (Truty & Urrutia, 2007). For example, TGFβ is a crucial driver of the activity of cancer-associated fibroblasts (CAFs) where it stimulates the deposition of extracellular matrix (Korc, 2007). TGFβ also promotes tumor immune evasion by inducing an immunosuppressive phenotype in myeloid cells and T cells (Fadok *et al*, 1998; Thomas & Massague, 2005; Principe *et al*, 2016).

TGFβ is a druggable target, inhibition of which has shown efficacy in multiple tumor types especially in combination with immune checkpoint blockade (Mariathasan *et al*, 2018; Tauriello *et al*, 2018). However, inhibition of the TGFβ signaling pathway in PDA has not been successful in preclinical or clinical studies (Hezel *et al*, 2012). This is due to the fact that canonical TGFβ signaling directly suppresses epithelial cancer cell growth in a context-dependent manner (Bardeesy *et al*, 2006; Ijichi *et al*,

1  Hamon Center for Therapeutic Oncology Research, University of Texas Southwestern Medical Center, Dallas, TX, USA
2  Cancer Biology Graduate Program, University of Texas Southwestern Medical Center, Dallas, TX, USA
3  Department of Biological Sciences, University of Texas at El Paso, El Paso, TX, USA
4  Division of Radiation Oncology, University of Texas MD Anderson Cancer Center, Houston, TX, USA
5  Department of Surgery, University of Texas Southwestern Medical Center, Dallas, TX, USA
6  Department of Molecular Biology, University of Texas Southwestern Medical Center, Dallas, TX, USA
7  Division of Hematology and Oncology, Department of Internal Medicine, University of Texas Southwestern Medical Center, Dallas, TX, USA
8  McDermott Center of Human Growth and Development, University of Texas Southwestern Medical Center, Dallas, TX, USA
9  Eli Lilly and Company, Indianapolis, IN, USA
   *Corresponding author. Tel: +1 214 648 5151; Fax: +1 214 648 4940; E-mail: rolf.brekken@utsouthwestern.edu

2006). Previously we reported that TGFβ signaling in stromal cells in PDA is a significant contributor to desmoplasia, immune suppression, and metastasis (Ostapoff *et al*, 2014). These data are particularly relevant given that tumor cell-specific deficiency in canonical TGFβ signaling via the loss of TGFβR2 or the downstream protein SMAD4 is common (up to 60%) in human PDA (Hahn *et al*, 1996; Waddell *et al*, 2015). Thus, TGFβ-mutant PDA is likely not subject to the tumor cell growth-suppressive activity of TGFβ yet maintains TGFβ-driven tumor promotion via the activity of TGFβ on stromal cells. Here, we investigate the effect of selective inhibition of stromal TGFβ signaling in PDA that harbors epithelial mutations in the canonical TGFβ pathway.

# Results

### Inhibition of TGFβR2 reduces stromal IL-6 production and tumor cell STAT3 activation

In earlier studies, we employed human PDA xenografts that metastasized robustly, and found that pharmacologic blockade of mouse TGFβR2 with 2G8, a rat anti-mouse TGFβR2 monoclonal antibody, resulted in striking reductions in metastatic spread (Ostapoff *et al*, 2014). These models enabled us to specifically inhibit stromal TGFβ signaling without blocking TGFβR2 signaling in the human tumor cells. Those results strongly support that targeting stromal TGFβ signaling can have antitumor effects through inhibiting a stromal paracrine signaling network. To better understand this stromal paracrine network, we performed a mouse-specific qPCR array study (Fig 1A) and an ELISA-based (Quansys) mouse chemokine assay with the xenografts (Fig 1B and Appendix Fig S1) and found that mouse interleukin 6 (IL-6) was consistently downregulated in the human xenografts after treatment with 2G8. We then investigated the effect of 2G8 on IL-6 secretion in two different genetically engineered mouse models (GEMMs), *KIC* ($Kras^{LSL-G12D/+}$; $Cdkn2a^{flox/flox}$; $Ptf1a^{Cre/+}$) and *KPC* ($Kras^{LSL-G12D/+}$; $Trp53^{LSL-R172H/+}$; $Ptf1a^{Cre/+}$) mice. We first validated the on-target efficacy of 2G8 by evaluating the activation of SMAD2. We found that in *KIC* and *KPC* tumors, 2G8 significantly reduced the SMAD2 activation (Fig 1E, H and I). Furthermore, we confirmed that the effect of 2G8 on IL-6 secretion was not specific to xenografts, as each GEMM treated with 2G8 showed a reduction in IL-6 (Fig 1C and D, and Appendix Fig S2).

Previous studies have shown that IL-6 is required to promote pancreatic intraepithelial neoplasia progression and PDA

development through the activation of STAT3 (Corcoran *et al*, 2011; Lesina *et al*, 2011; Zhang *et al*, 2013). Moreover, increased epithelial STAT3 activity correlates with decreased survival and advanced tumor stage in PDA patients (Nagathihalli *et al*, 2015). Therefore, we investigated the expression of IL-6 receptor (IL-6R) by immunohistochemistry in *KIC* and *KPC* mice and found that IL-6R was expressed robustly in cancer cells (Fig 1G). We evaluated the level of phosphorylated STAT3 after 2G8 treatment and found that 2G8 significantly reduced epithelial STAT3 activation in the GEMMs (Fig 1F, J and K). This suggests that TGFβ signaling promotes the secretion of IL-6 from stromal cells, which then induces STAT3 activation in PDA cancer cells.

### CAFs are the major source of IL-6 regulated by TGFβ in PDA

To identify the stromal cell type that secretes IL-6 in a TGFβ-dependent manner, we performed single-cell RNA sequencing (scRNA-seq) using whole tissue samples derived from normal mouse pancreas, early PDA, and late PDA from *KIC* mice (Hosein *et al*, 2019). We found that TGFβR1/R2 was mainly expressed by fibroblasts in the normal pancreas as well as early PDA (Fig 2A). In contrast, in late PDA, cancer cells and stromal cells expressed TGFβR1/R2. We validated the scRNA-seq results by Western blotting using primary mouse PDA cell lines (mPLRB8 and mPLRB9 from *KIC* mice, KPC-M01, KPC-M09 from *KPC* mice, BMFA3, CT1BA5 from *KPfC* ($Kras^{LSL-G12D}$; $Trp53^{flox/flox}$; $Pdx^{Cre/+}$)), macrophages (monocytes, M1- and M2-like), and fibroblasts (NIH3T3 and pancreatic stellate cells (PSC) (Fig 2B). These data suggest that a wide variety of cell types might be regulated by TGFβ in PDA. IL-6 in comparison was expressed by fibroblasts in normal pancreas and early lesions, while in more advanced PDA, fibroblasts and macrophages contributed to IL-6 production (Fig 2A). Consistently, in advanced PDA, fibroblast population-1 that expressed higher levels of TGFβR2 also expressed higher levels of IL-6. Based on this observation, we queried data from the Cancer Genome Atlas (TCGA) and found that there was a positive, statistically significant correlation between the expression of *IL6* and *TGFBR2* in human PDA (Fig 2C).

To determine whether fibroblasts or macrophages or both produced IL-6 in a TGFβ-dependent manner, we investigated the response to TGFβ stimulation. Mouse macrophages (monocytes, M1 and M2) and fibroblasts (NIH 3T3) were stimulated with TGFβ +/− 2G8, and active SMAD2 was evaluated (Fig 3A). TGFβ stimulated canonical signaling that was sensitive to 2G8 treatment in each cell type. To determine whether IL-6 was secreted in a TGFβ-dependent

---

**Figure 1. Inhibition of stromal TGFβR2 reduces IL-6 production and tumor cell STAT3 activation in PDA.**

A   Mouse qPCR array analysis was performed with Colo357 and MiaPaca-2 orthotopic tumor samples treated with saline (control) or 2G8 (*n* = 3/group) to detect changes in mouse stromal gene expression after treatment. A heat map was generated with ratios of gene expression (2G8 versus control).

B   ELISA-based Quansys mouse chemokine assay was performed with Capan-1 (Cap), MiaPaCa-2 (Mia), Colo357 (Colo), and C5LM2 (C5) orthotopic tumor samples treated with saline (control) or 2G8 (*n* = 3/group) to detect mouse chemokine changes after treatment. Change in mouse IL-6 was shown.

C, D   *KIC* mice were treated for 4 weeks, and *KPC* mice were treated for 55 days with Mac84 (control) or 2G8. Tumors from were collected for mouse IL-6 ELISA (*n* = 6/group).

E–K   *KIC* mice were treated for 4 weeks, and *KPC* mice were treated for 55 days with Mac84 (control) or 2G8. The activation of SMAD2 (P-Ser465/467) (E and H–I) and STAT3 (P-Tyr705) (F and J–K) and expression of IL-6R (G) were detected by immunohistochemistry (*n* = 4/group). Scale bars outside the magnification boxes = 50 μm, and scale bars inside the magnification boxes = 10 μm.

Data information: All data are reported as mean ± SD. *P* values versus control by *t*-test are indicated.

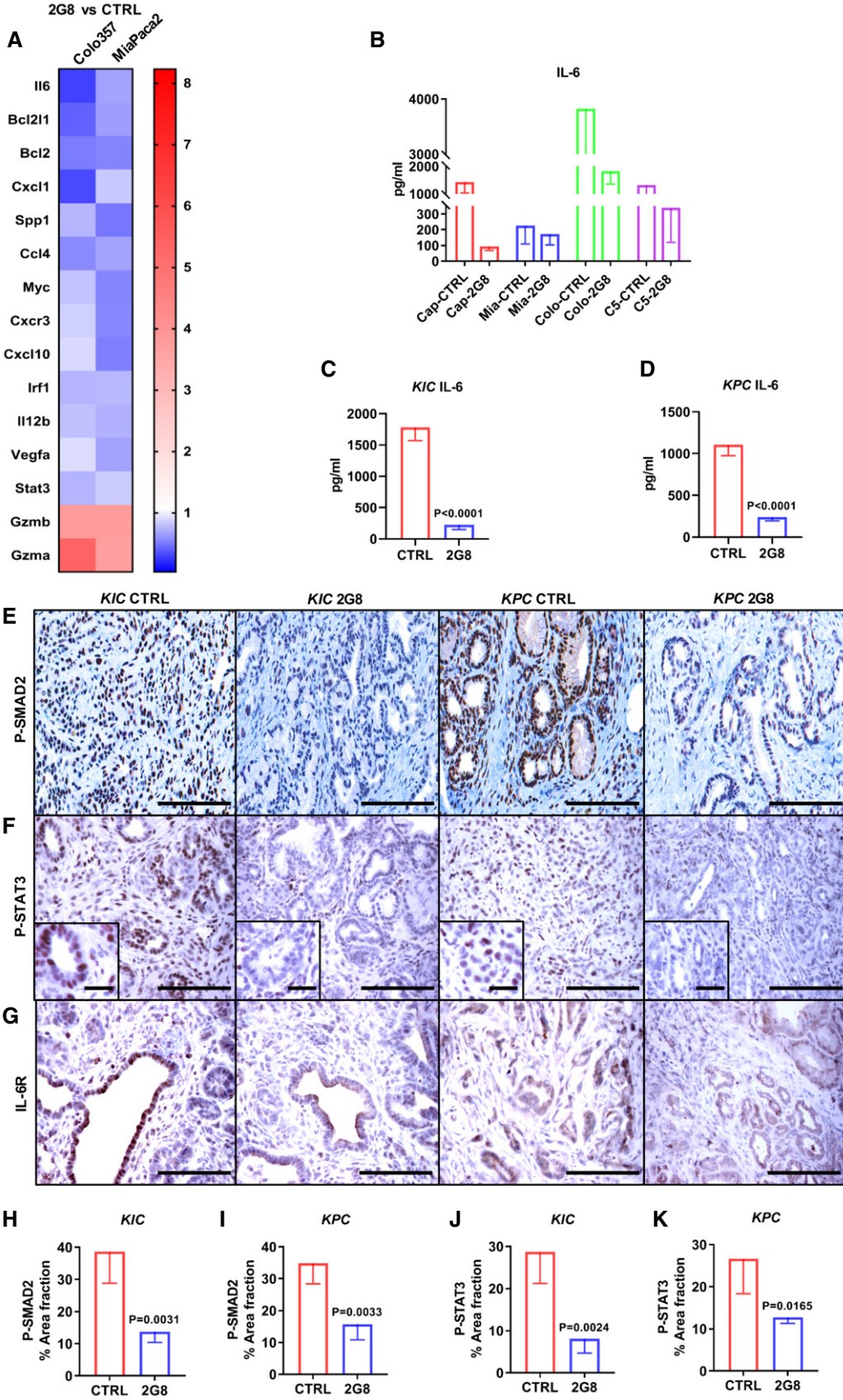

Figure 1.

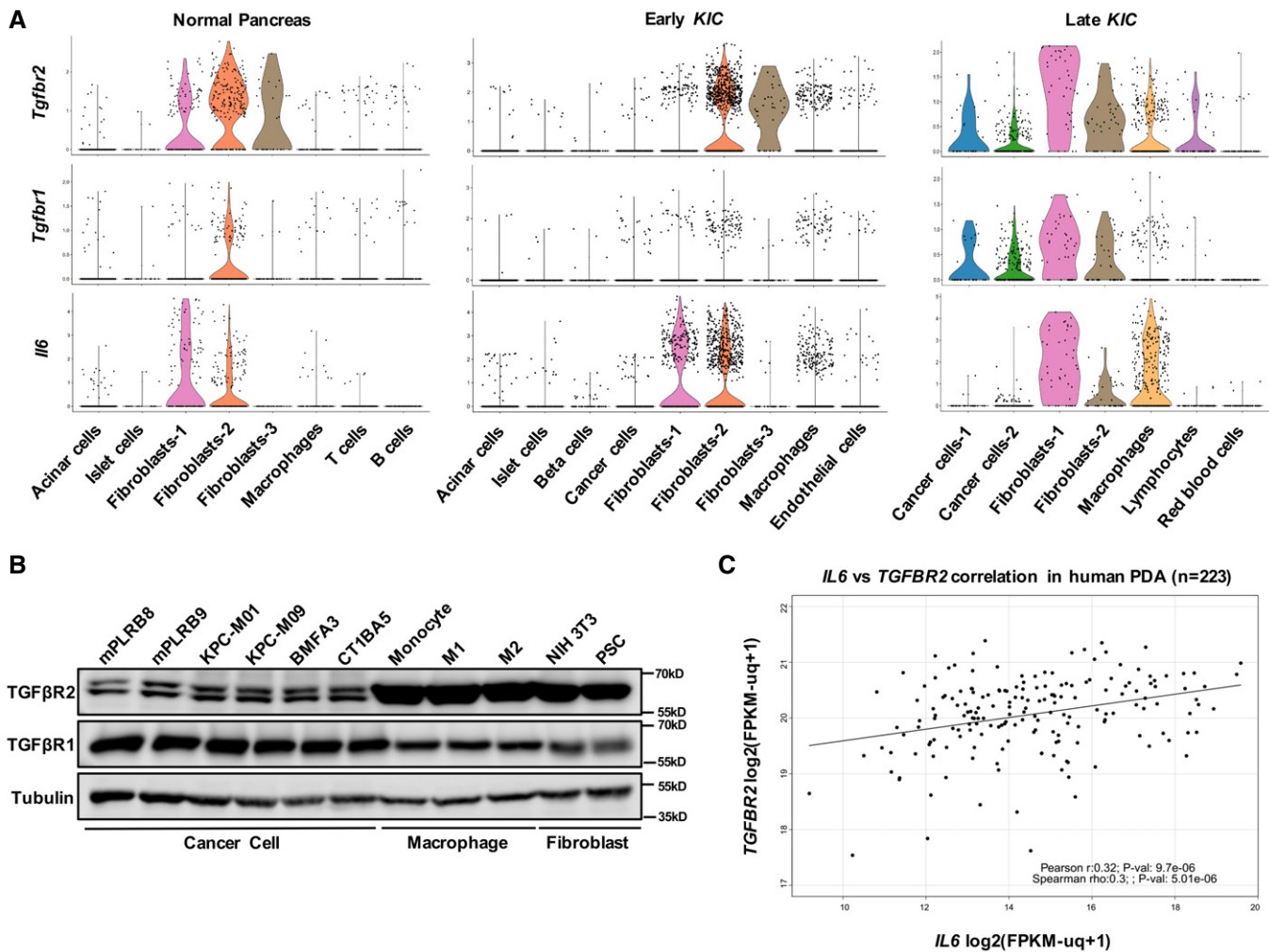

**Figure 2. CAFs are the major source of IL-6 in PDA.**

A Single-cell RNA sequencing was performed to profile cell populations in normal mouse pancreas (n = 2), early KIC (40-day-old, n = 2), and late KIC (60-day-old, n = 3) pancreata. Samples from the same stage were pooled. Violin plots of expression of Il6, Tgfbr1, and Tgfbr2 in distinct cell populations is shown.

B The expression of TGFβR1 and TGFβR2 in cell lysates harvested from KIC (mPLRB8, mPLRB9), KPC (KPC-M01, KPC-M09), and KPfC (BMFA3, CT1BA5) mouse cancer cells, mouse macrophages (RAW 264.7), and mouse fibroblasts (NIH 3T3 and pancreatic stellate cells). RAW 264.7 cells were induced into M1 (30 ng/ml LPS for 18 h) or M2 (20 ng/ml IL-4 for 18 h) macrophages. Tubulin was used as a loading control.

C Pearson and Spearman correlation of the expression of IL6 and TGFBR2 in PDA patients from TCGA (n = 223) using R.

Source data are available online for this figure.

manner, conditioned media was evaluated by ELISA. We found that, as reported (Martinez *et al*, 2008), M1-type macrophages secreted IL-6, but did so in a TGFβ-independent manner (Fig 3B). In contrast, TGFβ induced a significant increase in IL-6 production by fibroblasts (Fig 3C). We also found that TGFβ induced fibroblast secretion of leukemia inhibitory factor (LIF), an IL-6 family member (Appendix Fig S3A). Moreover, the induction of IL-6 and LIF could be blocked by 2G8. These data provide an explanation for the reduction of IL-6 caused by 2G8 treatment observed *in vivo*.

Recently, several studies, including ours, suggest the existence of two different subtypes of CAFs in PDA (Ohlund *et al*, 2017; Bernard *et al*, 2019; Elyada *et al*, 2019; Hosein *et al*, 2019). Of the two CAF populations, one appears to be inflammatory, characterized by

PDGFRα expression and the secretion of a variety of inflammatory chemokines. This population is referred to the fibroblast population-1 (Fig 2A). The other population is characterized by αSMA expression and typically considered a myofibroblast. A recent study reports that the inflammatory phenotype of CAFs is mediated by IL-1, while TGFβ promotes a myofibroblast phenotype (Biffi *et al*, 2019). We investigated the effect of TGFβ and IL-1 on IL-6 secretion in NIH 3T3, PSC, and two human pancreatic CAF cell lines (CAF-PC1 and CAF-PC2). Of the two human CAF cell lines, CAF-PC1 expressed the myofibroblast marker αSMA, while CAF-PC2 expressed the inflammatory CAF marker PDGFRα (Appendix Fig S3B). In NIH 3T3, PSC, and CAF-PC2, we found that TGFβ and IL-1 induced IL-6 secretion. In addition, the two cytokines had combinatory effect on IL-6 induction (Fig 3D–F). Interestingly, we found

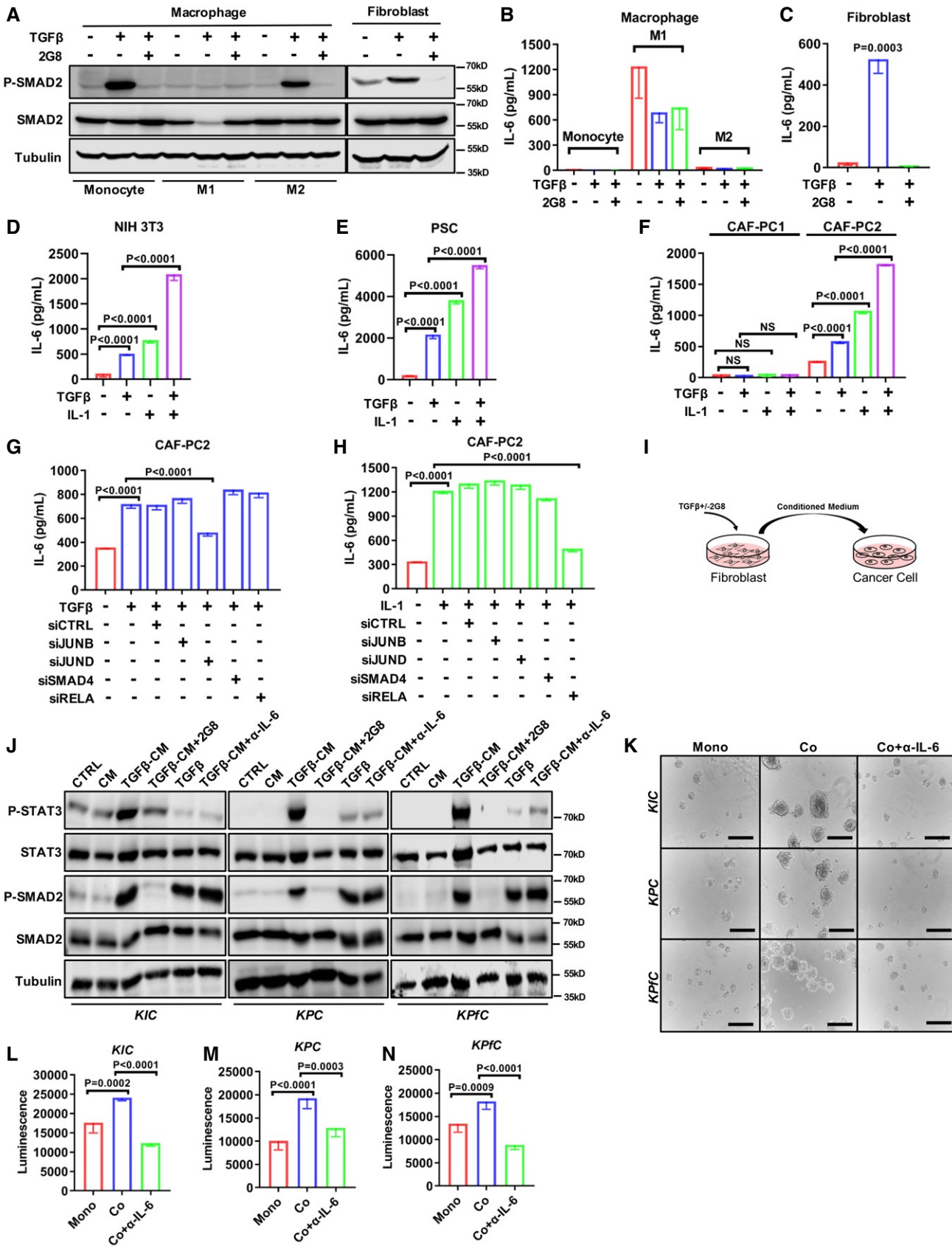

Figure 3.

◄

**Figure 3.** CAF-secreted IL-6 is regulated by TGFβ and activates STAT3 in pancreatic cancer cells.

A–C Control, M1 (LPS stimulated), and M2 (IL-4 stimulated) RAW 264.7 cells, and NIH 3T3 cells were treated with TGFβ (30 ng/ml), or TGFβ + 2G8 (100 ng/ml) for 24 h. Cell lysates were harvested and blotted for P-SMAD2 (P-Ser465/467), SMAD2, and tubulin (A). Conditioned media (CM) was collected for mouse IL-6 ELISA (B–C). P value by ANOVA is shown.

D–F NIH 3T3 (D), pancreatic stellate cells (PSC) (E), and human CAF cell lines CAF-PC1 and CAF-PC2 (F) were treated with TGFβ (30 ng/ml) and/or IL-1α (1 ng/ml) for 24 h. CM was collected for mouse or human IL-6 ELISA. P values by t-test are shown.

G, H CAF-PC2 cells were treated with TGFβ (30 ng/ml) (G) or IL-1α (1 ng/ml) (H) for 24 h with siRNA-mediated knockdown of JUNB, JUND, SMAD4, or RELA. CM was collected and subjected to human IL-6 ELISA. P values by t-test were shown.

I, J KIC (mPLRB9), KPC (KPC-M09), and KPfC (BMFA3) cell lines were treated with normal DMEM (CTRL), CM from NIH 3T3 (CM), CM from TGFβ-treated NIH 3T3 (TGFβ-CM), CM from TGFβ-treated NIH 3T3 + 2G8 (TGFβ-CM + 2G8) (I), normal DMEM + TGFβ (TGFβ), and CM from TGFβ-treated NIH 3T3 + IL-6 neutralizing antibody (TGFβ-CM + IL-6 Ab). Cell lysates were harvested and blotted for P-STAT3 (P-Tyr705), STAT3, P-SMAD2 (P-Ser465/467), SMAD2, and tubulin (J).

K–N 3D culture: cells were seeded on poly-HEMA-coated 96-well plates and cultured for 4 days (5,000 cancer cells for monoculture, 3,000 cancer cells + 2,000 NIH 3T3 for co-culture). IL-6 neutralizing antibody (100 ng/ml). Scale bars = 50 μm. n = 5/group, P values by t-test are shown.

Data information: All data are reported as mean ± SD.
Source data are available online for this figure.

CAF-PC1, which had myofibroblast features, did not secrete IL-6 in response to TGFβ or IL-1 stimulation (Fig 3F).

To identify the transcription factors that mediated the IL-6 secretion upon TGFβ treatment, we knocked down transcription factors that mediate the canonical (SMAD4) or non-canonical (JUNB, JUND) TGFβ signaling in CAF-PC2 (Zhang, 2009; Massague, 2012). We found that the IL-6 induction was dependent on JUND (Fig 3G). In comparison, we found that IL-1 induced IL-6 in an NF-κB-dependent manner (Fig 3H), consistent with prior results (Biffi et al, 2019). In summary, these data suggest that TGFβ promotes IL-6 secretion in different types of cytokine-secreting fibroblasts through non-canonical TGFβ signaling.

### TGFβ-induced IL-6 from fibroblasts activates STAT3 in PDA cancer cells and promotes tumorigenesis

To investigate the effect of IL-6 secreted by fibroblasts on cancer cells, conditioned media from mouse fibroblasts treated with TGFβ (TGFβ-CM) was collected (Fig 3I). Cancer cells (mPLRB9 from KIC, KPC-M09 from KPC and BMFA3 from KPfC) were incubated with TGFβ-CM and subjected to Western blotting for STAT3 signaling (Fig 3J). Compared to control media (CTRL) and fibroblast conditioned media without TGFβ pretreatment (CM), TGFβ-CM activated STAT3 in cancer cells, an effect that was efficiently inhibited by 2G8 treatment of the fibroblasts (TGFβ-CM + 2G8). TGFβ also activated canonical TGFβ signaling in cancer cells. Importantly, the treatment of cancer cells with TGFβ directly had mild effect on STAT3 activation. Furthermore, the activation of STAT3 by TGFβ-CM was sensitive to IL-6 blockade (TGFβ-CM + α-IL-6). These data indicate that TGFβ-induced IL-6 from fibroblasts can directly activate STAT3 in PDA cancer cells.

As illustrated by Lesina et al (2011), Zhang et al (2013), IL-6 is required during PDA progression, and we have demonstrated that fibroblasts are a major source of IL-6 in the tumor microenvironment. To understand the function of fibroblast-secreted IL-6 during PDA progression, a 3D co-culture study to recapitulate the tumorigenesis process in vitro was performed (Fig 3K). In comparison with cancer cell monoculture, the co-culture grew significantly faster and larger in the presence of fibroblasts (Fig 3L–N). Furthermore, such growth was inhibited by neutralizing IL-6 in the co-culture. This highlights the direct effect of IL-6 on promoting tumor progression. During tumor progression, epithelial–mesenchymal transition (EMT) is a biological program often associated with

advanced tumors. It is characterized by the loss of epithelial cell markers and the gain of mesenchymal features (Kalluri & Weinberg, 2009). Through EMT, epithelial cancer cells often become more invasive and resistant to therapy. TGFβ is a known driver of EMT (Xu et al, 2009); thus, we investigated if IL-6 contributed to TGFβ-induced EMT. As expected, TGFβ induced a decrease in epithelial marker (E-cadherin) and an increase in mesenchymal marker (N-cadherin) in mPLRB9 and KPC-M09 cells; however, IL-6 did not induce any noticeable EMT changes or alter the effect of TGFβ (Appendix Fig S4).

### CAF-secreted IL-6 inhibits NK cell activity and leads to PDA metastasis

We have identified that TGFβ induces a paracrine IL-6 signal from CAFs that stimulates STAT3 activity in PDA cells and promotes 3D growth in vitro. However, how CAFs affect the tumor microenvironment through this paracrine signal is still unknown. To better understand the stromal changes after 2G8 treatment, we exploited the human tumor xenografts (MiaPaca-2 and Colo357) by performing RNA sequencing using mouse-specific primers. We found a distinct stromal gene signature change with 2G8 treatment (Fig 4A). We then performed Ingenuity Pathway Analysis (IPA) on the stromal gene signature and found that the altered pathways were closely associated with immune functions (Fig 4B and C).

PDA is considered to be an immunologically "cold" tumor, which is consistent with the observation that PDA is typically poorly responsive to immune therapy. TGFβ is an immunosuppressive cytokine that has been shown to facilitate tumor immune evasion in multiple tumor types (Mariathasan et al, 2018; Tauriello et al, 2018). Consistently, we previously reported that inhibition of mouse TGFβR2 with 2G8 had immune effects in human tumor xenografts (Ostapoff et al, 2014). Importantly, these xenografts were grown in NOD SCID mice, which lack cells of the adaptive immune system (Hudson et al, 1998). Thus, the immune-associated changes suggested an activation of the innate immune system. Specifically, we found the expression of multiple cytotoxic genes (Gzma, Gzmb, and Prf1) were enhanced by 2G8 (Fig 4D–F). These changes suggested that 2G8 induced a change in the activation of natural killer (NK) cells.

Multiple studies indicate that TGFβ can directly inhibit NK cell activation (Rook et al, 1986; Zhong et al, 2010; Rouce et al, 2016; Viel et al, 2016). We were interested in determining if IL-6 also

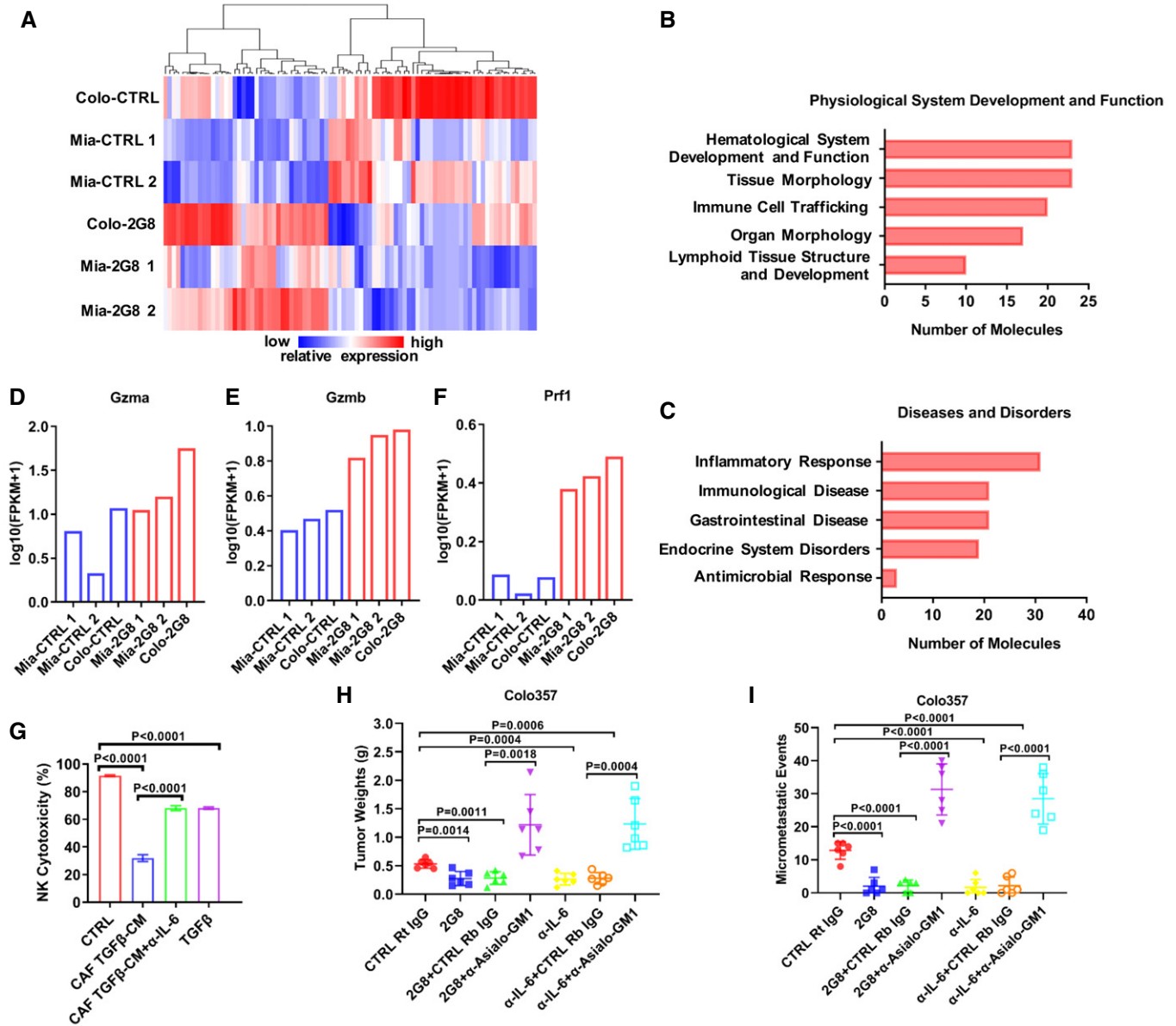

**Figure 4. TGFβ-induced IL-6 from CAFs inhibits NK cell activity and promotes pancreatic cancer.**

A–F Saline and 2G8-treated MiaPaca-2 and Colo375 xenografts were subjected to RNA-seq analysis for mouse gene expression changes. A heat map was generated with gene clustering (A). IPA showing pathways most effected by 2G8 (B–C). Molecules most significantly effected by 2G8 treatment. *Y*-axis values shown are log10 (FPKM+1) of the transcript levels (D-F).

G An *in vitro* NK cell cytotoxicity assay was performed. Human NKL cells were used as effector cells, and mouse pancreatic cancer BMFA3 cells were used as target cells in normal DMEM (control), conditioned medium collected from TGFβ (30 ng/ml)-treated human pancreatic CAFs (CAF-PC2; CAF TGFβ-CM), CAF TGFβ-CM + IL-6 antibody (100 ng/ml), or normal DMEM + TGFβ (TGFβ). Living cells were labeled with CFSE, and dead cells were labeled with 7-AAD. Samples were analyzed by flow cytometry. Cytotoxicity percentage was calculated using the formula (7-AAD-positive cells %)/(7-AAD-positive cells % + CFSE-positive cells %) × 100%. *n* = 4/group, *P* values by *t*-test are shown.

H, I Human pancreatic cancer cell line Colo357 was orthotopically implanted into NOD SCID mice. After tumor establishment, mice were randomized to receive rat IgG Mac84 (control), 2G8, or anti-mouse IL-6 antibody MP5-20F3 (each 30 mg/kg 2×/week, *n* = 6/group) for 3 weeks. For NK cell depletion, prior to therapies, mice received 50 μg of control rabbit IgG or anti-Asialo-GM1 3 days in a row. For maintenance, 25 μg of control rabbit IgG or anti-Asialo-GM1 was given twice a week throughout the whole study. Tumors were harvested for analysis, and metastatic burden was determined by histologic evaluation of H&E-stained liver tissue. Ten sections of the anterior lobe of the liver (*n* = 6 per group) were scored for lesions. *P* values by *t*-test are shown.

Data information: Data in (G–I) are reported as mean ± SD.

contributed to NK cell function. We performed an *in vitro* NK cell killing assay to test this hypothesis. We found that in the presence of TGFβ-CM, NK cell activity was significantly reduced, and it could be partially rescued by neutralizing IL-6 (Fig 4G). This suggests that TGFβ has direct and indirect (through IL-6) effects on NK cell antitumor activity. To validate these findings, we performed an *in vivo*

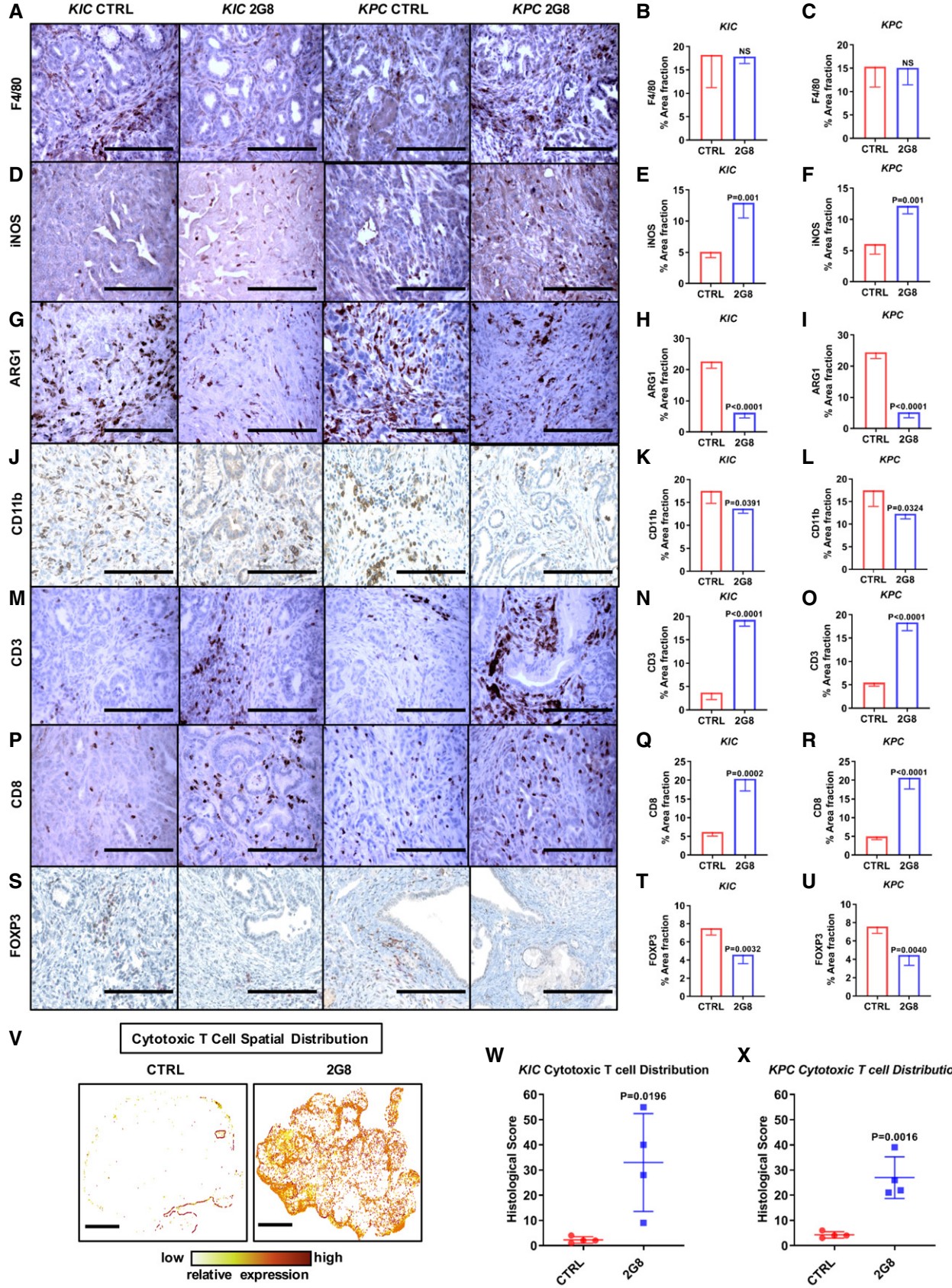

**Figure 5.**

**Figure 5. Inhibition of stromal TGFβR2 reverses immunosuppression in PDA.**

A–U    *KIC* mice were treated for 4 weeks, and *KPC* mice were treated for 55 days with Mac84 (control) or 2G8. Immune landscape changes were detected by
        immunohistochemistry for macrophage makers F4/80 (total, A–C), iNOS (M1, D–F), ARG1 (M2, G–I), CD11b (myeloid-derived suppressor cells, J–L), and T-cell markers
        CD3 (M–O), CD8 (P–R), and FOXP3 (regulatory T cells, S–U) in *KIC* and *KPC* mice. Scale bars = 50 μm. *n* = 4/group, *P* values by *t*-test are shown.

V–X    Spatial distribution analysis of CD8-positive T cells after 2G8 treatment. Details are described in Appendix Fig S7. Sample images of *KPC* tumors are shown (V).
        Substantial cytotoxic T-cell infiltration was observed after 2G8 treatment. Histological score was calculated using the following formula (W, X): low IHC cells
        % + 2× (medium IHC cells %) + 3× (high IHC cells %). Scale bars = 500 μm. *n* = 4/group, *P* values versus control by *t*-test are shown.

Data information: All data are reported as mean ± SD.

study (Fig 4H and I). Consistent with our previous study (Ostapoff *et al*, 2014), 2G8 reduced xenograft progression in NOD SCID mice; however, the antitumor effects of 2G8 were lost when we ablated NK cells using the NK cell-depleting antibody α-Asialo-GM1 (Appendix Fig S5). To determine the contribution of IL-6 *in vivo*, we performed the same experiment using a neutralizing IL-6 antibody (MP5-20F3). The inhibition of IL-6 curtailed tumor progression similar to 2G8 in NOD SCID mice but the effect was also lost with NK cell ablation. These results were recapitulated in NSG animals that lack NK cells (Appendix Fig S6). Taken together, these data indicate that TGFβ induces the secretion of IL-6 from CAFs to inhibit the NK cell activity, and inhibition of stromal TGFβ signaling or IL-6 rescues NK cell activity, resulting in reduced PDA progression.

### TGFβR2 blockade alters the immune microenvironment but does not increase survival in PDA that has intact TGFβ signaling

We have found that in a xenograft setting, stromal TGFβR2 blockade enhances NK cell activity. To extend these studies, we investigated the effect of 2G8 therapy in immunocompetent GEMMs. Importantly, in contrast to xenografts, in GEMMs 2G8 inhibits TGFβR2 on stromal cells and tumor cells, which is more relevant to the clinical application of the therapy. Immunohistochemical analysis of *KIC* and *KPC* tumors after 2G8 therapy supports that TGFβR2 blockade alters the immune microenvironment to favor antitumor immune activity (Fig 5A–U). This was shown by an increase in immune-stimulatory iNOS⁺ macrophages (Fig 5D–F), a decrease in immunosuppressive ARG1⁺ macrophages (Fig 5G–I), a decrease in CD11b⁺ myeloid-derived suppressor cells (Fig 5J–L), a significant increase in the infiltration of CD3⁺CD8⁺ cytotoxic T cells (Fig 5M–R and V–X, and Appendix Fig S7), and a decrease in FOXP3⁺ regulatory T cells in 2G8-treated animals (Fig 5S–U).

In addition, we found that 2G8 reduced STAT3 activation (Fig 1D–F). However, 2G8 therapy did not provide a survival benefit; in fact, it decreased survival (Fig 6A and B) consistent with studies from multiple groups. Given that the *KIC* and *KPC* GEMMs have intact TGFβ signaling, we evaluated tumor cell proliferation. We found that 2G8 stimulated epithelial cancer cell proliferation marked by elevated Ki67 (Fig 6C). This phenotype is similar to a prior study, in which we treated the *Kras-p53^{flox/+}* mice with a monoclonal antibody against TGFβ (Hezel *et al*, 2012). These results underscore the dual effects of canonical TGFβ signaling blockade in PDA and the difficulties of its therapeutic application.

### Loss-of-function mutations in *TGFBR2* are common in PDA patients

Given that the challenges associated with TGFβ blockade are likely due to the loss of a TGFβ-mediated suppression of tumor cell

proliferation, which is mediated via canonical TGFβ signaling in tumor cells (Bardeesy *et al*, 2006; Ijichi *et al*, 2006), we hypothesized that patients who harbor a loss-of-function mutation in *TGFBR2* could benefit from TGFβR2 inhibition.

To explore this, we investigated the percentages of PDA patients who have loss-of-function mutations in *TGFBR2*. The TCGA and UT Southwestern (UTSW) databases demonstrate that such mutations are found in up to 7% of the PDA patients (Fig 6D). Furthermore, we found that patients who harbored a *TGFBR2* mutation tended to have a worse prognosis (Fig 6E). To mimic this population of patients and understand the responsiveness of TGFβR2-mutant tumor cells to TGFβ, we generated two TGFβR2-deficient pancreatic cancer cell lines *Tgfbr2^{mut1}* and *Tgfbr2^{mut2}*. These lines were derived using CRISPR to remove *Tgfbr2* from the primary mouse cancer cell line (BMFA3), which was obtained from a *KPfC* animal. We found that TGFβ stimulation failed to activate canonical (P-SMAD2), or non-canonical (P-ERK1/2 and P-P38) TGFβ signaling in the TGFβR2-deficient cancer cells (*Tgfbr2^{mut1}* and *Tgfbr2^{mut2}*) compared to the control cell line (*Tgfbr2^{wt}*) (Fig 6F). Moreover, TGFβ did not induce SMAD4 nuclear translocation or EMT in the mutant cells (Fig 6G). Thus, the TGFβR2-deficient cells are unresponsive to TGFβ *in vitro*.

### TGFβR2 blockade has therapeutic efficacy in PDA that harbors a loss-of-function mutation in *Tgfbr2*

To test the hypothesis that PDA harboring a loss-of-function mutation in *Tgfbr2* is sensitive to TGFβR2 inhibition, we first investigated the effect of TGFβ on cell proliferation *in vitro*. As expected, TGFβ treatment inhibited the proliferation of *Tgfbr2^{wt}* control cells (Fig 7A); however, the *Tgfbr2^{mut1/2}* cell lines were not affected by TGFβ (Fig 7B and C). To extend these observations, we subcutaneously injected syngeneic mice with the parental and TGFβR2-mutant tumor cells and treated the mice with 2G8. We found that in the control tumors, 2G8 significantly increased tumor growth (Fig 7D), but 2G8 significantly slowed the growth of *Tgfbr2*-deficient tumors (Fig 7E and F).

We then performed a survival study by orthotopically implanting the control or mutant cells into syngeneic mice. Similar to the effect seen in the *KIC* model, 2G8 decreased the survival of the mice carrying the *Tgfbr2* wild-type tumors (Fig 7G and I) and enhanced tumor weight (Fig 7K). However, in the *Tgfbr2*-deficient tumors, 2G8 enhanced the survival and reduced tumor weight (Fig 7H, J and L). Immunohistochemical analysis demonstrated that 2G8 stimulated cancer cell proliferation in wild-type tumors but reduced tumor cell proliferation in *Tgfbr2*-deficient tumors (Fig 7M–O). Additionally, STAT3 activation was reduced in wild-type and *Tgfbr2*-deficient tumors consistent with blockade of TGFβ-mediated IL-6 secretion from CAFs (Fig 7P–R). Furthermore,

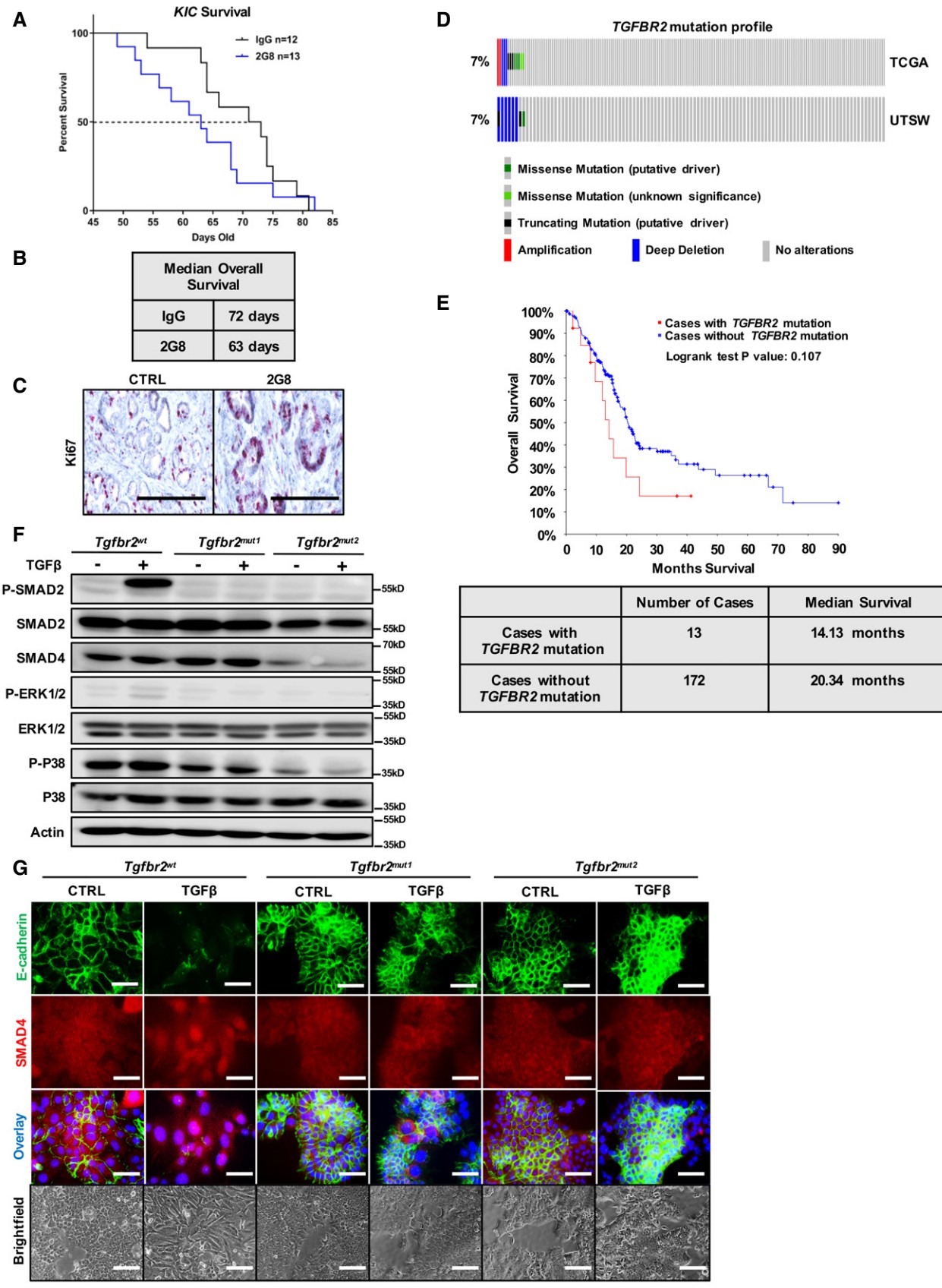

Figure 6.

**Figure 6. Loss-of-function mutations in *TGFBR2* are common in PDA.**

A, B  *KIC* mice were treated in a survival study with rat IgG Mac48 (control) or 2G8. Median overall survival of rat IgG treatment was 72 days, while median overall survival of 2G8 treatment was 63 days (B).

C  Tissues from (A) were stained for Ki67. Scale bars = 50 μm.

D  Mutation profile of *TGFBR2* in PDA patient samples from two independent sources (UTSW and TCGA).

E  Overall survival of PDA patients from TCGA with and without *TGFBR2* mutation.

F  Loss-of-function mutation of *Tgfbr2* was generated by CRISPR with two different gRNAs (mut1 and mut2) in the mouse PDA cell line BMFA3 derived from *KPfC* model. Control cell line (*Tgfbr2$^{wt}$*) and two *Tgfbr2*-mutant cell lines (*Tgfbr2$^{mut1}$* and *Tgfbr2$^{mut2}$*) were treated with TGFβ (30 ng/ml) for 5 h. Cell lysates were harvested and Western blotting for P-SMAD2, SMAD2, SMAD4, P-ERK1/2, ERK1/2, P-P38, P38, and actin was performed.

G  Immunofluorescence was performed to study the expression and localization of E-cadherin and SMAD4 after TGFβ treatment (30 ng/ml for 24 h) in control cell line and two *Tgfbr2*-mutant cell lines. Scale bars = 50 μm (fluorescent), scale bars = 100 μm (brightfield).

Source data are available online for this figure.

2G8 resulted in pronounced changes in the immune landscape in wild-type and *Tgfbr2*-deficient tumors marked by an infiltration of NK cells and T cells (Fig 7S–X). These results strongly suggest that PDA that harbors a loss-of-function mutation in *TGFBR2* will benefit from inhibition of stromal TGFβR2 signaling.

### Elucidation of the contribution of IL-6 signaling in the efficacy of stromal TGFβR2 blockade in TGFβR2-mutant PDA

We have identified that inhibition of the TGFβ-IL-6 paracrine signaling axis reduces STAT3 activation in cancer cells and also blunts immune evasion. To ascertain the contribution of IL-6 signaling on cancer cells to this effect, we knocked down IL-6 receptor (IL6RA) in the TGFβR2-mutant cells (Fig 8A) and evaluated the efficacy of TGFβR2 and IL-6 inhibition *in vitro* and *in vivo*. We first found that knocking down IL6RA significantly reduced the growth of 3D coculture *in vitro* and tumors *in vivo* (Fig 8B–D). Additionally, Ki67 expression and STAT3 activation were reduced (Fig 8G, H and K and L). This suggests that the direct effect of IL-6 on cancer cells via STAT3 activation is important during the progression of TGFβR2-mutant tumors.

Further, we found that inhibition of TGFβR2 or IL-6 had additive therapeutic efficacy only *in vivo* but not *in vitro* when IL6RA was knocked down in cancer cells (Fig 8B–D). This strongly suggests that the inhibition of the TGFβ-IL-6 paracrine signal affects the activity of other stromal cells *in vivo*, which are not present in the 3D coculture system, and results in additive antitumor effects. Since we have demonstrated the inhibitory effect of TGFβ-IL-6 signal on NK cell activity, and we observed a significant increase in NK cells in IL6RA knockdown tumors treated with 2G8 or anti-IL-6 antibody (Fig 8I and N), we hypothesized that NK cells are the main stromal cell type mediating the additional antitumor effects. To test this idea, we ablated NK cells in mice bearing IL6RA knockdown tumors and found that 2G8 and the anti-IL-6 antibody lost efficacy in the absence of NK cells (Fig 8E and F). Taken together, these data demonstrate that the therapeutic efficacy of TGFβR2 blockade in TGFβR2-mutant PDA is a result of the inhibition of the CAF TGFβ-IL-6 paracrine signal that drives cancer cell proliferation and inhibits NK cell activity (Appendix Fig S8).

## Discussion

In PDA, TGFβ has been reported to contribute to tumor progression (Friess *et al*, 1993; Massague, 2008). However, neither pharmacological nor genetic ablation of TGFβ signaling can derail the progression of the tumor, due to the suppressive effects of the cytokine on cancer cells (Bardeesy *et al*, 2006; Ijichi *et al*, 2006; Hezel *et al*, 2012). In contrast, TGFβ also modulates many microenvironmental events that cancer cells exploit, including CAF activity and immune regulation (Ostapoff *et al*, 2014; Principe *et al*, 2016). As a result, the output of a response to TGFβ is context-dependent and clinical trials targeting the TGFβ signaling pathway in PDA have not been successful. Previously, we identified that selective blockade of stromal TGFβR2 resulted in reduced metastasis, a more epithelial differentiated tumor cell phenotype, and immune changes consistent with innate immune activation (Ostapoff *et al*, 2014). Moreover, other groups also have demonstrated that TGFβ drives microenvironmental aspects of PDA progression, including alteration of stromal and hematopoietic cell function that promote tumor aggressiveness (Principe *et al*, 2016).

We hypothesized that a TGFβ-driven stromal paracrine network existed in PDA, and therefore, by using species-specific arrays in xenograft models, we identified IL-6 as the most consistent stromal factor elevated by TGFβ. Furthermore, we found that CAFs are a major source of TGFβ-induced IL-6 expression. Several recent studies including ours demonstrate the existence of fibroblast heterogeneity within PDA (Ohlund *et al*, 2017; Bernard *et al*, 2019; Elyada *et al*, 2019; Hosein *et al*, 2019). Two major populations of CAFs have been identified, one is characterized by an inflammatory feature, while the other has been known as myofibroblast. One recent study reports that IL-1 induces inflammatory CAFs that secret a wide variety of cytokines including IL-6 and LIF, while TGFβ antagonizes this process by downregulating IL1R1 expression, promoting the myofibroblast feature (Biffi *et al*, 2019). Congruent with this, we also identified NF-κB as a major mediator of IL-1 induction of IL-6 secretion (Fig 3H). However, in contrast, we found that TGFβ induced IL-6 secretion by fibroblasts in a JUND-dependent manner (Fig 3D–G). Moreover, we also found that TGFβ had combinatory effect with IL-1 on IL-6 induction. Additionally, in our scRNA-seq study, we observed that the inflammatory CAF population was characterized by a high level of *Tgfbr1/2* and *Il6* expression (Fig 2A). In PDA patients, the expression of *IL6* positively correlated with *TGFBR2* (Fig 2C). Moreover, induction of IL-6 and LIF by TGFβ has been reported in different types of fibroblasts including CAFs in other cancers (Eickelberg *et al*, 1999; Albrengues *et al*, 2014; Shintani *et al*, 2016; Dufour *et al*, 2018; Curtis *et al*, 2019). In addition, consistent with our findings, JUND was previously reported to mediate TGFβ-induced IL-6 secretion in human lung fibroblasts (Eickelberg *et al*, 1999). Interestingly, we also

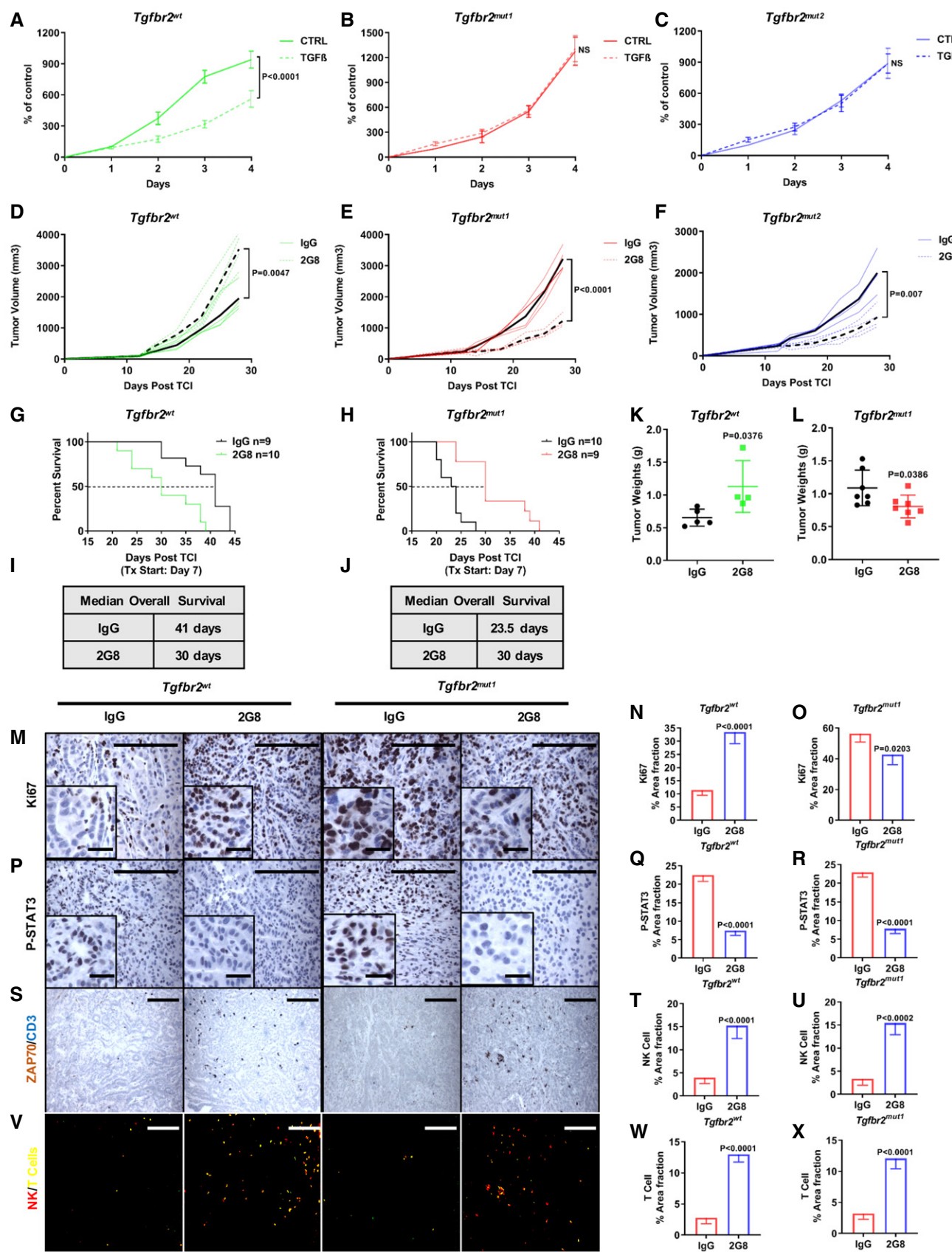

**Figure 7.**

**Figure 7. TGFβR2 blockade has therapeutic efficacy in TGFβR2-mutant PDA due to the inhibition of stromal TGFβ signaling.**

A–C  Control cell line (*Tgfbr2^wt*) (A) and two *Tgfbr2*-mutant cell lines (*Tgfbr2^mut1* and *Tgfbr2^mut2*) (B, C) were treated with TGFβ at different time points, and cell growth (MTT) assays were performed. *n* = 8, *P* value by *t*-test is shown. NS, not significant.

D–F  Subcutaneous tumors established from control cell line (D) and two *Tgfbr2*-mutant cell lines (E, F) in C57Bl/6 mice received rat IgG Mac48 (control) or 2G8 (each 30 mg/kg 2×/week, *n* = 4/group). Therapy started at day 12 post-tumor cell injection; mice were on therapy for 16 days. Tumor volume was measured twice per week. *P* values by *t*-test are shown.

G–L  Control cell line (*Tgfbr2^wt*; G and I) and the *Tgfbr2*-mutant cell line (*Tgfbr2^mut1*; H and J) were orthotopically implanted in C57Bl/6 mice. Mice received rat IgG Mac48 or 2G8, *n* = 9–10/group for survival study (each 30 mg/kg 2×/week). Mice were sacrificed when they became moribund. Tumors were harvested and weighed (K–L). *P* values by *t*-test are indicated.

M–X  Immunohistochemistry with tumor samples from (G, H) for Ki67 (M–O), P-STAT3 (P–R), ZAP70, and CD3 (S–X) was performed. ZAP70 is a common marker for NK cells and T cells and was stained with a brown chromogen, and CD3 is a specific T-cell marker and was stained with a blue chromogen. Brown signal was converted to red signal, and blue signal was converted to green by ImageJ (V); therefore, NK cells are highlighted as red (ZAP70⁺CD3⁻) and T cells are highlighted as yellow (ZAP70⁺CD3⁺). Scale bars outside the magnification boxes = 50 μm, scale bars inside the magnification boxes = 10 μm. *n* = 4/group, *P* values versus control by *t*-test are shown.

Data information: All data are reported as mean ± SD.

found that myofibroblasts did not induce IL-6 in response to TGFβ or IL-1 (Fig 3F), supporting that this CAF population is tumor suppressive (Ozdemir *et al*, 2014; Rhim *et al*, 2014). In summary, these findings underscore the complexity of CAF biology in PDA and the need for further studies on the TGFβ signaling pathways in different populations of CAFs.

As a major downstream signaling mediator of IL-6, the activity of STAT3 has been shown to have important functions during PDA development. For example, the genetic ablation of IL-6 or inactivation of STAT3 substantially inhibits the initiation and progression of PDA (Corcoran *et al*, 2011; Lesina *et al*, 2011; Zhang *et al*, 2013). A more recent study shows that in PDA, CAFs induce an invasive EMT and proliferative phenotype of cancer cells through the activation of STAT3 in a co-culture system (Ligorio *et al*, 2019). Since the major source of IL-6 is the tumor stroma, STAT3 signaling represents a signaling pathway that is driven by the stroma during the PDA progression. We demonstrated that 2G8 treatment significantly reduced the activity of STAT3 in PDA cancer cells *in vitro* and *in vivo* by downregulating stromal IL-6, and therefore, TGFβ is a potential target, the blocking of which can interrupt this stromal tumorigenic signal.

Besides the effects in cancer cells, STAT3 signaling has also been reported to inhibit the function of immune cells, resulting in the immune suppressive tumor microenvironment. Tumor-infiltrating immune cells, including myeloid cells and lymphocytes, display a constitutive activation of STAT3 due to the stromal

paracrine signals in the tumor niche (Yu *et al*, 2009). Many studies have demonstrated the inhibitory effects of STAT3 signaling on NK cell function. For example, in a syngeneic bladder cancer mouse model, NK depletion abrogates the effects of a STAT3 inhibitor on tumor rejection (Kortylewski *et al*, 2005). In another study, conditional *Stat3* knockout in NK cells in a transgenic mouse model showed elevated granzyme B and perforin levels under physiological conditions (Gotthardt *et al*, 2014). In addition, metastatic lung nodules are greatly reduced in mice with STAT3-deficient NK cells when challenged with B16-F10 melanoma cells. Moreover, the growth of tumors formed by *v-abl*-transformed cells can be inhibited in the mice with NK cell-specific *Stat3* knockout and STAT3 inhibition can prolong the survival of the mice bearing the *v-abl*-transformed tumors. Given the inhibitory effects of STAT3 signaling on NK cells, we found that CAFs in PDA were a major contributor to the inhibition of NK cell function due to IL-6 secretion, resulting in the loss of innate immune antitumor effects. Consistently, we noticed that blocking TGFβR2 or IL-6 significantly reduced stromal STAT3 activation (Fig 8H and L), while increasing NK cells in syngeneic models (Fig 8I and N). TGFβ is a well-studied inhibitor of effector lymphocytes including NK cells; therefore, by inhibiting stromal TGFβR2 there is a combinatory effect that enhances the function of NK cells due to a direct inhibition of TGFβ signaling on immune cells and also due to a reduction of IL-6. We have also shown that TGFβR2 blockade resulted in a substantial change in the immune landscape of PDA including a significant infiltration of cytotoxic T cells (Figs 5 and 7V–X, and 8J

**Figure 8. Inhibition of the TGFβ-IL-6 paracrine signal on cancer cells and NK cells results in the therapeutic efficacy in TGFβR2-mutant PDA results.**

A  Control shRNA or two different shRNAs against IL6RA were used to knock down IL6RA in the *Tgfbr2*-mutant cell line *Tgfbr2^mut1*. Cell lysates were harvested and Western blotting for IL6RA and tubulin.

B, C  3D culture: Control or IL6RA knockdown *Tgfbr2^mut1* cells were seeded on poly-HEMA-coated 96-well plates and cultured for 4 days (5,000 cancer cells for monoculture, 3,000 cancer cells + 2,000 NIH 3T3 for co-culture). Control IgG, 2G8, and IL-6 neutralizing antibody (each 100 ng/ml, *n* = 5/group). Scale bars = 50 μm. *P* values by *t*-test are shown (C).

D–F  Subcutaneous tumors established from control and IL6RA knockdown *Tgfbr2*-mutant cell lines in C57Bl/6 mice received rat IgG Mac48 (control), 2G8, or anti-mouse IL-6 antibody MP5-20F3 (each 30 mg/kg 2×/week, *n* = 5/group). For NK cell depletion, prior to therapies, mice received 50 μg of control rabbit IgG or anti-Asialo-GM1 3 days in a row. For maintenance, 25 μg of control rabbit IgG or anti-Asialo-GM1 was given twice a week throughout the whole study. Therapy started at day 12 post-tumor cell injection, and mice were on therapy for 16 days. Tumor volume was measured twice per week. *P* values by *t*-test are indicated.

G–N  Immunohistochemistry with tumor samples from (D) for Ki67 (G and K), P-STAT3 (H and L), ZAP70, and CD3 (I–J and M–N) was performed. NK cells are highlighted as red (ZAP70⁺CD3⁻), and T cells are highlighted as yellow (ZAP70⁺CD3⁺) by ImageJ (N). Scale bars outside the magnification boxes = 50 μm, scale bars inside the magnification boxes = 10 μm. *n* = 5/group, *P* values versus control by *t*-test are shown.

Data information: All data are reported as mean ± SD.
Source data are available online for this figure.

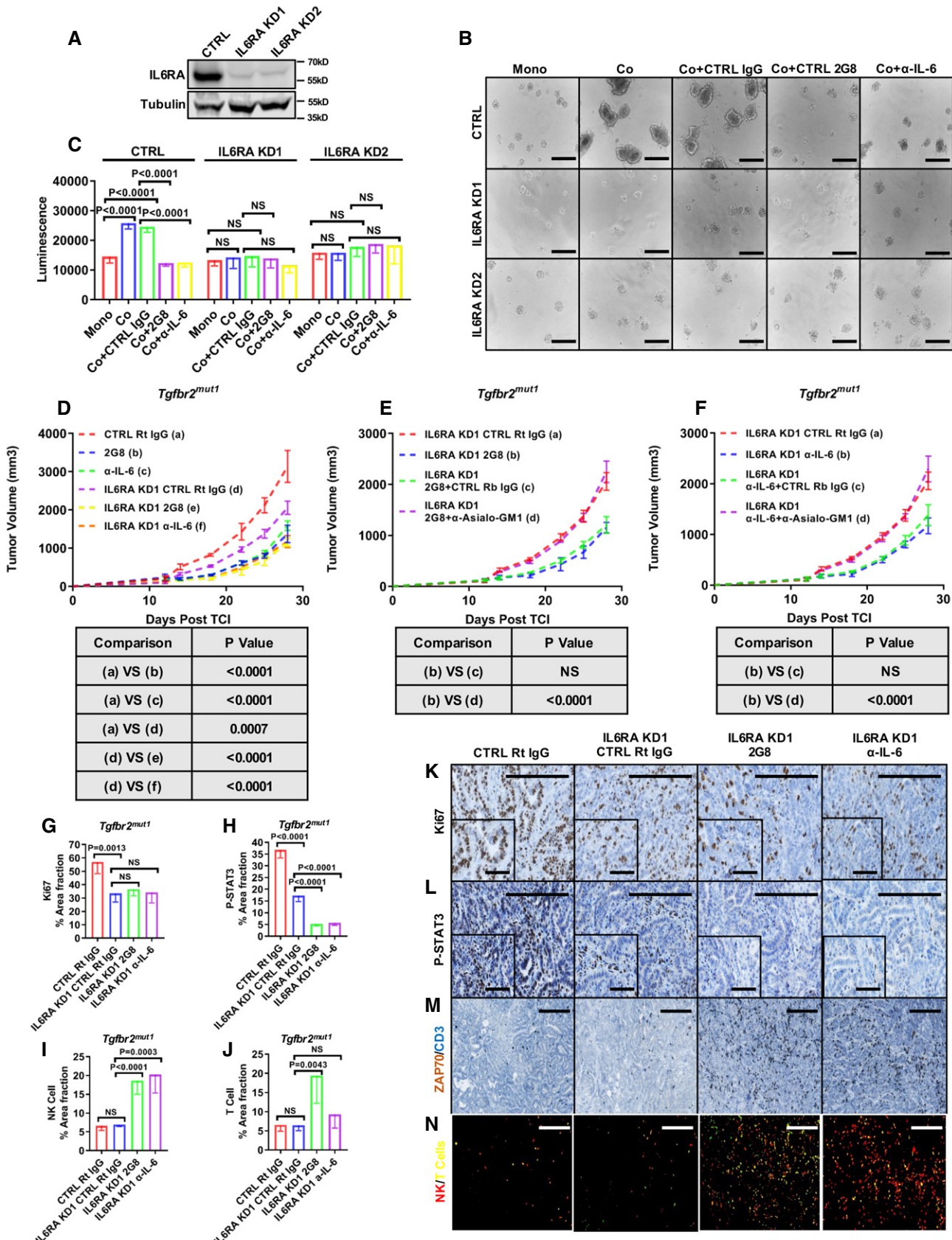

**Figure 8.**

and N). However, we found that the infiltration of T cells was not major contributor to the antitumor effects observed (Fig 8E and F). This could be due to T-cell exhaustion or a lack of effective neoantigens in this model. Nonetheless, blockade of TGFβ creates a favorable environment for immune therapies.

Despite the antitumor effects due to the inhibition of the stromal TGFβ signaling, global TGFβR2 blockade resulted in a worse outcome in tumors with wild-type TGFβR2. This was due to increased tumor cell proliferation in the presence of TGFβR2 blockade. In contrast, if TGFβR2 is absent on cancer cells, this negative effect of TGFβ blockade is lost. Therefore, the outcome of TGFβR2 blockade is a net inhibition of stromal TGFβ signaling and therapeutic efficacy. Up to 60% of PDA patients have loss-of-function *TGFBR2* or *SMAD4* mutations (Waddell *et al*, 2015), and importantly, these patients have significantly shorter survival and a worse prognosis after surgery (Tascilar *et al*, 2001). Based on our study, we suggest that in these patients, TGFβ blockade will have therapeutic benefit. TGFβ is an attractive therapeutic target for enhancing the effects of chemotherapy and immune therapy. Multiple therapeutic strategies targeting the TGFβ pathway are undergoing clinical studies in different cancers including PDA (de Gramont *et al*, 2017; Yingling *et al*, 2018). Our study underscores the importance of stratifying the patients in these studies such that those with tumor cell mutations in TGFβ signaling would be candidates for inhibition of stromal TGFβ signaling in combination with standard or immune therapy.

# Materials and Methods

### Cell lines

Human pancreatic cancer cell lines Capan-1 and MiaPaca-2 were obtained from ATCC, and Colo357 cells were obtained from Dr. Jason Fleming (H. Lee Moffitt Cancer Center, Tampa, FL). C5LM2 was a cell line derived from liver metastasis from a Panc-1 tumor bearing mouse isolated by our laboratory. Human CAF cell lines PC1 and PC2 were obtained from Dr. Martin Fernandez-Zapico (Mayo Clinic, Rochester, MN). An NKL cell line was obtained from Dr. Chengcheng Zhang (UTSW Medical Center, Dallas, TX). Mouse primary pancreatic cancer cell lines (Ostapoff *et al*, 2014) and stellate cells (Ohlund *et al*, 2017) were isolated as described previously. KPC-M01 and KPC-M09 were derived from *KPC* mice, mPLRB8 and mPLRB9 were derived from *KIC* mice, and BMFA3 and CT1BA5 were derived from *KPfC* mice. RAW 264.7 cells and NIH 3T3 cells were obtained from ATCC. Human cell lines were authenticated to confirm origin, and all cell lines were confirmed to be free of mycoplasma (e-Myco kit, Boca Scientific) before use. Cells were cultured in DMEM (Invitrogen) or RPMI (Invitrogen) containing 10% FBS and maintained at 37°C in an atmosphere of 5% $CO_2$.

### Reagents and antibodies

Anti-P-STAT3 (Tyr705, #9145, dilution 1:1,000 for Western blot), anti-STAT3 (#4904, dilution 1:1,000 for Western blot), anti-P-SMAD2 (Ser465/467, #3108, dilution 1:1,000 for Western blot), anti-SMAD2 (#3103, dilution 1:1,000 for Western blot), anti-E-cadherin (#3195, dilution 1:1,000 for Western blot, 1:200 for immunofluorescence),

anti-N-cadherin (#14215, dilution 1:1,000 for Western blot), anti-vimentin (#5741, dilution 1:1,000 for Western blot), anti-ARG1 (#93668), anti-CD8α (#98941), anti-P-ERK1/2 (Thr202/Tyr204, #4370, dilution 1:1,000 for Western blot), anti-ERK1/2 (#9102, dilution 1:1,000 for Western blot), anti-P-P38 MAPK (Thr180/Tyr182, #4511, dilution 1:1,000 for Western blot), anti-P38 MAPK (#9212, dilution 1:1,000 for Western blot), anti-PDGFRα (#3174, dilution 1:1,000 for Western blot), and anti-ZAP70 (#2705) antibodies were obtained from Cell Signaling Technology. Anti-TGFβR2 (sc-17791, dilution 1:1,000 for Western blot) and anti-SMAD4 (sc-7966, dilution 1:1,000 for Western blot, 1:100 for immunofluorescence) antibodies were obtained from Santa Cruz Biotechnology. Anti-IL-6Rα (AF1830, dilution 1:1,000 for Western blot) antibody and human IL-6 neutralizing antibody (MAB2061) were obtained from R&D Systems. Anti-tubulin (MCA77D800, dilution 1:2,000 for Western blot) antibody was obtained from Bio-Rad Laboratories. Anti-iNOS (PA1-21054) and anti-CD3 (PA1-29547) antibodies were obtained from Thermo Fisher Scientific. Anti-F4/80 (NBP2-12506) antibody was obtained from Novus Biologicals. Anti-Ki67 (ab15580) and anti-TGFβR1 (ab31013, dilution 1:1,000 for Western blot) antibodies were obtained from Abcam. Anti-αSMA (001, dilution 1:1,000 for Western blot) antibody was obtained from Biocare Medical. Mouse IL-6 neutralizing antibody (mabg-mil6-3) was obtained from InvivoGen. Recombinant human TGFβ1 (100-21C) was obtained from PeproTech, Inc. Anti-CD335 (137604) antibody and recombinant mouse IL-6 (575704) were obtained from BioLegend. Mouse (400-ML-005/CF) and human (200-LA-002/CF) IL-1α, mouse IL-6 (DY406), mouse LIF (DY449), and human IL-6 (DY206) DuoSet ELISA kits were obtained from R&D Systems.

### 3D co-culture and cell viability assay

The 3D co-culture assay was performed by coating 96-well plates with poly-2-hydroxyethyl methacrylate (poly-Hema; P3932, Sigma-Aldrich). For monoculture, 5,000 cancer cells and for co-culture, 2,000 cancer cells and 3,000 fibroblasts were seeded per well. Cells were incubated for 7 days until spheroid formation. Cell viability was measured using CellTiter-Glo (G7570, Promega).

For MTT assay, cells were plated at 5000 cells per well in 96-well plates and incubated with or without TGFβ (30 ng/ml) for up to 96 h. By 24, 48, 72, and 96 h, 3-(4,5-dimethylthiazol-2-yl)-2,5-diphenyltetrazolium bromide (M6494, Thermo Fisher Scientific) was added to the cells and the cells were incubated for 4 h. Dimethyl sulfoxide (DMSO, 472301, Sigma-Aldrich) was then added to the wells and kept at room temperature in the dark for 1 h. Absorbance at 570 nm was recorded to measure cell viability.

### NK cell cytotoxicity assay

The NK cell cytotoxicity assay was performed following the instruction of the basic cytotoxicity assay kit (969, ImmunoChemistry Technologies). In brief, target cells (BMFA3) were incubated with effector cells (NKL) under different conditions. Samples and control were stained with CFSE labeling viable cells and 7-AAD labeling dead cells. Cell percentages were analyzed by flow cytometry. The cytotoxicity percentage was calculated using the formula (7-AAD-positive cells %)/(7-AAD-positive cells % + CFSE-positive cells %) × 100%.

## Western blot analysis

Cells were extracted with radioimmunoprecipitation assay buffer (50 mM Tris–HCl, pH 8.0, 150 mM NaCl, 0.1% SDS, 0.5% sodium deoxycholate, and 1% Nonidet P-40) for SDS–PAGE. Protein was determined using a BCA Protein Assay Kit (23225, Thermo Fisher Scientific). Laemmli sample buffer was added to the protein lysates and boiled for 10 min. The proteins were then resolved by SDS–PAGE, electrophoretically transferred to nitrocellulose membranes, and blocked in 5% nonfat dry milk or bovine serum albumin. Blocking buffer was then removed, and the membranes were incubated with primary antibody in TBST (10 mM Tris–HCl, pH 7.5, 150 mM NaCl, 0.05% Tween 20) for 1 h, then washed $3 \times 10$ min with TBST, and incubated with horseradish peroxidase-conjugated anti-mouse or rabbit secondary antibody (Jackson ImmunoResearch Laboratories) for 1 h. The secondary antibody was removed by washing $3 \times 10$ min with TBST. The membranes were incubated with SuperSignal West Pico substrate (34580, Thermo Fisher Scientific) for the detection of the immunoreactive bands.

## ELISA

For ELISA arrays, assays were performed with the tumor lysates following the instructions of the mouse cytokine screen (Quansys Biosciences) or MILLIPLEX mouse cytokine/chemokine panel (Millipore). For IL-6 and LIF ELISA, assays were performed following the instructions of the DuoSet ELISA kits (R&D Systems). In summary, 96-well plates were incubated with capture antibody overnight at room temperature, washed with wash buffer, blocked with reagent diluent for 1 h, and washed again. Then, 100 μl of sample was added per well to the plates and incubated for 2 h at room temperature. The plates were washed, and 100 μl of detection antibody was added per well and incubated for 2 h at room temperature. Then, the plates were further washed and 100 μl of substrate solution was added to each well and incubated for 20 min at room temperature. Afterward, 50 μl of stop solution per well was used to stop the reaction and the absorbance at 450 nm was measured.

## RNA isolation, qPCR array, and RNA sequencing

For RNA isolation, tumor tissues were harvested in RLT lysis buffer and purified according to instructions of the RNeasy Plus Kit (QIAGEN). An Agilent 2100 Bioanalyzer was used to determine RNA quality, and only samples with an RNA integrity number score of 7 or higher were used. The samples were subjected to qPCR array or RNA sequencing. For qPCR array, experiments were performed using mouse RT$^2$ Profiler PCR Array (QIAGEN) following the kit protocol (target genes were listed in Appendix Table S1, additional information is available through https://www.qiagen.com/). For RNA sequencing, RNA concentration was determined by a Qubit fluorometer (Thermo Fisher Scientific) and then prepared with TruSeq Stranded Total RNA Sample Prep Kit (Illumina). Samples were quantified, normalized, and sequenced on the Illumina HiSeq 2500 with at least 25 million reads per sample. FASTQ files were aligned to mouse mm10 reference transcriptome. A differential expression analysis was performed using edgeR, and statistical cutoffs of FDR $\leq 0.01$, log2CPM $\geq 0$ and FC cutoffs of $-1.5 \geq FC \geq 1.5$ were used to identify statistically significant transcripts. Data analysis was performed using an IPA tool (Qiagen). A heat map was clustered with top differentially expressed genes by hierarchical clustering using R.

For scRNA-seq (Hosein et al, 2019), normal mouse pancreas, 40-day-old KIC (early KIC), and 60-day-old KIC pancreases (late KIC) were freshly isolated and enzymatically digested. A library was generated with the single-cell suspension using the 10× Chromium system (10× Genomics, Inc.). Cells were loaded with Single Cell 3′ Gel Beads into a Single Cell A Chip and run on the Chromium Controller. Gel beads in emulsion were incubated, broken, and then cleaned up by Silane magnetic beads. During the incubation, Read 1 primer sequence was added and barcoded cDNA was amplified by PCR after the cleanup. Sample size was determined on the Agilent Tapestation 4200 using the DNAHS 5000 tape, and concentration was determined by the Qubit Fluorometer. Samples were then enzymatically fragmented and went through size selection. A Read 2 primer sequence, sample index, and both Illumina adapter sequences were added during library preparation. Samples were then cleaned up by Ampure XP beads, and quality control was performed by the DNA 1000 tape on the Agilent Tapestation 4200. Sequencing depth was more than $10^5$ reads per cell.

Cell Ranger version 1.3.1 (10× Genomics) was used to process the raw sequencing data. Raw BCL files were converted to FASTQ files and aligned to mouse mm10 reference transcriptome. Transcript counts of each cell were quantified using barcoded UMI and 10xBC sequences. The gene × cell expression matrices were loaded to the R-package Seurat version 2.3.0 for downstream analyses. Cells with low quality were filtered out based on the number of genes detected, UMI, and mitochondrial gene content. By regressing out the number of UMI and percentage of mitochondrial gene content, the data were scaled and dimensional reduction was performed with principle component analysis and visualization using tSNE plots. Cell clusters were identified via a FindClusters function with a resolution of 0.6, and a likelihood ratio-based test or an area under the curve-based scoring algorithm (Seurat package) was used to identify marker genes for each cluster, and the expression levels of known marker genes in each cell cluster were validated.

## CRISPR knockout and knockdown

Oligos of the gRNAs were annealed with T4 polynucleotide kinase (New England Biolabs) by PCR. Annealed oligos were then ligated to PX458 vector with FastDigest BbsI (FD1014, Thermo Fisher Scientific) and T7 DNA Ligase (M0318, New England Biolabs) by PCR. Mixture from the reaction was then transformed into the NEB 5-α Competent E. coli (High Efficiency). Plasmids were extracted from the expanded colonies and sent to the UTSW sequencing core for sequencing. Plasmids with correct gRNA sequences or empty vector control were transfected into BMFA3 cells with Lipofectamine 2000 (11668027, Thermo Fisher Scientific). Positive cells expressing GFP were sorted as single clones and expanded. Each expanded clone was subjected to validation through PCR and functional study by testing the response to the TGFβ treatment.

**The paper explained**

**Problem**

TGFβ is a druggable target, inhibition of which has shown efficacy in multiple tumor types especially in combination with immune checkpoint blockade. However, inhibition of the TGFβ signaling pathway in pancreatic cancer (PDA) has not been successful in preclinical or clinical studies. This is due to the fact that canonical TGFβ signaling directly suppresses epithelial cancer cell growth in a context-dependent manner. Previously we reported that TGFβ signaling in stromal cells in PDA is a significant contributor to desmoplasia, immune suppression, and metastasis. Importantly, in human PDA tumor cell-specific deficiency in canonical TGFβ signaling via the loss of TGFβR2 or the downstream protein SMAD4 is common (up to 60%). Thus, we hypothesized that TGFβ-mutant PDA is not subject to the tumor cell growth-suppressive activity of TGFβ yet TGFβ activation of stromal cells still drives tumor progression.

**Results**

We demonstrate that TGFβ induces cancer-associated fibroblasts to produce IL-6 that promotes tumorigenesis via STAT3 activation in PDA cells and also inhibits NK cell activity. Thus, TGFβR2 blockade *in vivo* inhibits STAT3 activation in PDA cells and alters the immune microenvironment. However, TGFβR2 blockade has therapeutic efficacy only in PDA that harbors tumor cell loss-of-function mutations in *Tgfbr2*. We further demonstrate that the efficacy of TGFβR2 blockade in TGFβR2-mutant tumors is due to the inhibition of TGFβ signaling in stromal cells and IL-6 signaling in cancer cells and NK cells.

**Impact**

Our study highlights the potential benefit of TGFβ blockade in PDA and the importance of stratifying PDA patients who might benefit from such therapy.

Oligos used for the cloning of gRNAs (mut1 and mut2) targeting *Tgfbr2* are

mut1 5′-CACCGTCCACAGGACGATATGCAG-3′
 5′-AAACCTGCATATCGTCCTGTGGAC-3′
mut2 5′-CACCGTCACCCGACTTGGGAACGTG-3′
 5′-AAACCACGTTCCCAAGTCGGGTGAC-3′

Primers for PCR validation are

mut1 5′-CTGAGAGGGCGAGGAGTAAAG-3′
 5′-CCACTCACTCACCCGACTTG-3′
mut2 5′-GCTGCATATCGTCCTGTGG-3′
 5′-AGTGGACTGTCCTGCTCTTTTG-3′

For siRNA-mediated knockdown experiments, cells were plated 18–24 h before transfection at an initial confluence of 60–80%. TransIT-siQUEST reagent and siRNAs were prepared and added according to manufacturer instructions (Mirus Bio LLC). siRNAs were added to the cells at a final siRNA complex concentration of 1 μM. All human siRNAs were purchased from Dharmacon (siGENOME). The shRNA constructs were as follows: IL6RA KD1 (TRCN0000068294, Sigma), IL6RA KD2 (TRCN0000375089, Sigma). All the shRNAs were in the vector pLKO.1-Puro. For a negative control, we used pLKO.1-puro carrying a sequence targeting EGFP (Sigma). Production of viral supernatants and infection of target cells have been described previously (Huang *et al*, 2016). 4 μg/ml puromycin was used for selection of the target cells.

## Animal studies

Mice were treated with 2G8 (provided by Eli Lilly and Company), a rat anti-mouse TGFβR2 IgG2a that does not target human TGFβR2 (Ostapoff *et al*, 2014), AFRC Mac 48 (Mac48), a rat IgG2a specific for phytochrome (European Collection of Animal Cell Cultures), anti-mouse IL-6 monoclonal antibody MP5-20F3 (Bio × Cell), control polyclonal rabbit IgG (Bio × Cell), and anti-Asialo-GM1, a rabbit polyclonal NK cell-depleting antibody (Wako Pure Chemicals).

### Genetic endpoint and survival model

*KIC* (40- to 43-day-old) and *KPC* (90-day-old) mice were randomized to receive Mac48 or 2G8 (all 30 mg/kg, 2×/week). For endpoint study, *KIC* mice were treated for 4 weeks and *KPC* mice were treated for 55 days. Mice were then sacrificed, and organs were harvested for analysis. For survival study, mice received therapies until being moribund and were sacrificed and organs were harvested for analysis.

### Xenograft model

Eight-week-old NOD SCID or NSG mice were orthotopically injected with $1 \times 10^6$ or $2 \times 10^6$ cells (Capan-1, MiaPaCa-2, Colo357, or C5LM2). Mice were randomized to receive saline, Mac48, 2G8, MP5-20F3 (all 30 mg/kg, 2×/week) 10 days after injection and treated for 3 weeks. For NK cell depletion, prior to therapies, mice received 50 μg of control rabbit IgG or anti-Asialo-GM1 3 days in a row. For maintenance, 25 μg of control rabbit IgG or anti-Asialo-GM1 was given 2×/week throughout the whole study. Mice were sacrificed 3.5 weeks after tumor cell injection. Tumors were harvested for analysis, and gross metastases or micrometastases were counted.

### Syngeneic model

BMFA3 cells were injected subcutaneously ($5 \times 10^5$ cells) or orthotopically ($2.5 \times 10^5$ cells) in 6- to 8-week-old C57BL/6 mice. For the subcutaneous model, 12 days after tumor cell injection, mice were randomized to receive Mac48, 2G8, or MP5-20F3 (all 30 mg/kg, 2×/week). For NK cell depletion, prior to therapies, mice received 50 μg of control rabbit IgG or anti-Asialo-GM1 3 days in a row. For maintenance, 25 μg of control rabbit IgG or anti-Asialo-GM1 was given 2×/week throughout the whole study. Tumor volume was measured twice per week. For the orthotopic model, 7 days after tumor cell injection, mice were randomized to receive Mac48 or 2G8 (all 30 mg/kg, 2×/week) until being moribund and were then sacrificed and organs were harvested for analysis.

## Immunohistochemistry

Formalin-fixed, paraffin-embedded tissues were cut in 5-μm sections. Sections were evaluated by immunohistochemical analysis following our previously reported protocol (Sorrelle *et al*, 2019) using antibodies specific for anti-P-STAT3 (Tyr705, #9145, Cell Signaling, dilution 1:500), anti-ARG1 (#93668, Cell Signaling, dilution 1:500), anti-CD8α (#98941, Cell Signaling, dilution 1:1,000), anti-ZAP70 (#2705, Cell Signaling, dilution 1:500), anti-P-SMAD2 (Ser465/467, AB3849, Millipore, dilution 1:500), anti-IL-6Rα (AF1830, R&D Systems, dilution 1:250), anti-iNOS (PA1-21054, Thermo Fisher Scientific, dilution 1:200), anti-CD3 (PA1-29547, Thermo Fisher Scientific, dilution 1:1,000), anti-F4/80 (NBP2-12506,

Novus Biologicals, 1:200), anti-FOXP3 (MAB8214, Novus Biologicals, dilution 1:500), anti-CD11b (ab133357, Abcam, dilution 1:1,000), and anti-Ki67 (ab15580, Abcam, dilution 1:1,000). Images were obtained with a Nikon Eclipse E600 microscope using a Niko Digital Dx1200me camera and ACT1 software (Universal Imaging Corporation). Pictures were analyzed using NIS Elements (Nikon).

### Spatial distribution analysis

mages were annotated using Aperio ImageScope (Leica Biosystems Imaging, Inc.). The annotated images were imported to the image analysis software, Definiens Architect 2.6. (Definiens, Inc.). First, we used the machine learning algorithm to isolate the IHC stain from the counterstain (Appendix Fig S6). Then, we ran a tissue segmentation algorithm to exclude the glass background. That was followed by using nuclear detection and cell simulation algorithms to segment and detect the IHC-positive and IHC-negative cells. To obtain the histological score, the IHC-positive cells were classified into low, medium, and high, based on the signal intensity. All results were exported in a.csv format. Using JMP-Pro (SAS Institute, Inc.), we plotted the x and y coordinates of the IHC-positive cells and density plots were created.

### Statistics

For each animal experiment, all the mice were purchased at the same time with similar age. For subcutaneous tumors, before treatment, mice were randomized into different groups according to the initial tumors sizes and the gender of the mice. Therefore, the average tumor sizes of all the groups were similar. For orthotopic tumor models, before treatment, mice were randomized after surgery according to gender. Laboratory personnel designed the experiment, randomized mice, and placed mice in cages based on which therapy was given. Animal resource center (ARC) staff were provided with drugs in blinded vials and asked to treat the appropriate cages. The ARC staff and the lab personnel consulted on animal health and made decisions regarding animals that needed to be sacrificed due to tumor or therapy-induced morbidity. Male and female mice were included in equal numbers for each animal experiment. Treatment group was also blinded during tumor size measurement. All data are reported as mean ± SD. Statistical analysis was performed with a 2-tailed $t$-test or ANOVA using GraphPad Prism software and R language (https://www.R-project.org/). $P < 0.05$ was considered statistically significant. All *in vitro* experiments were performed with at least three biological replicates.

### Study approval

All animals were housed in a pathogen-free facility with 24-h access to food and water. Animal experiments in this study were approved by and performed in accordance with the institutional animal care and use committee at the UTSW Medical Center at Dallas.

## Data availability

The datasets produced in this study are available in the following database: RNA-Seq data: Gene Expression Omnibus

GSE135578 (https://www.ncbi.nlm.nih.gov/geo/query/acc.cgi?acc =GSE135578).

**Expanded View** for this article is available online.

## Acknowledgements

This work was supported by a sponsored research agreement from Eli Lilly & Company (SRA5200886401 RAB), NIH grants R01 (CA192381) to RAB and U54 (CA210181 Project 2) to RAB and EJK, the Effie Marie Cain Fellowship to RAB and a predoctoral fellowship from the American Heart Association, and the Harry S. Moss Heart Trust (19PRE34380436) to ZW. We thank Drs. Jason Fleming, Martin Fernandez-Zapico, and Alec Zhang for provision of cell lines and reagents. We acknowledge helpful input from Dr. Jill Westcott, other members of the Brekken laboratory and Drs. Herb Zeh, Daolin Tang, and Michael Lotze and editorial assistance provided by Dave Primm of the UTSW Department of Surgery.

## Author contributions

HH: study concept and design, acquisition of data, analysis, and interpretation of data, drafting of the manuscript. YZ: acquisition of data. VG: acquisition of data, analysis, and interpretation of data. NS: acquisition of data. MMZ: bioinformatics analyses. JT: acquisition of data. WD: acquisition of data. SW: acquisition of data. MH: acquisition of data. ZW: acquisition of data and bioinformatics analyses. ANH: acquisition of data. AAS: acquisition of data and bioinformatics analyses. CX: bioinformatics analyses. EJK: bioinformatics analyses and interpretation of data. KED: study concept and design, analysis and interpretation of data. RAB: study concept and design, interpretation of data, and drafting of the manuscript.

## Conflict of interest

RAB reports receiving a commercial research grant from Eli Lilly & Company. KED is an employee of Eli Lilly and Company. The authors have no additional financial interests.

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
