## [Review Process File · EMBO Molecular Medicine]

Targeting TGF β R2-mutant tumors exposes vulnerabilities to stromal TGF β blockade in pancreatic cancer

Hucong Huang, Yuqing Zhang, Valerie Gallegos, Noah Sorrelle, Mohamed Medhat Zaid, Jason Toombs, Wenting Du, Steven Wright, Moriah Hagopian, Zhaoning Wang, Abdel Nasser Hosein, Adwait Amod Sathe, Chao Xing, Eugene J. Koay, Kyla E. Driscoll, Rolf A. Brekken

Review timeline:

Submission date:	25 February 2019
Editorial Decision:	28 March 2019
Revision received:	16 August 2019
Editorial Decision:	6 September 2019
Revision received:	11 September 2019
Accepted:	17 September 2019

Editor: Céline Carret

Transaction Report:

1st Editorial Decision

28 March 2019

Thank you for the submission of your manuscript to EMBO Molecular Medicine. We have now heard back from the three referees whom we asked to evaluate your manuscript.

You will see that they all find the topic of your manuscript interesting but they feel that the data need to be strengthened and they make constructive suggestions for that. Ref. 1 and 3 are more reserved than ref. 2 even though this referee shares concerns with the other reviewers. Overall, while the novelty is limited by existing knowledge, the paper could be published if more mechanism is provided. Ref. 3 suggests a list of experiments deemed essential for publication. Ref. 1 questions the choice of model, controls, would like to see better figures and statistical analyses, and discussion including on published literature. Ref. 2 requests a more detailed analysis on immune cells and like ref. 3 more mechanism. Upon cross-commenting, referees agree that given the limited novelty, the paper has to be strengthened by a more thorough and controlled mechanistic analysis. Recommendations should be followed to solidify the evidence for the present claims.

We would therefore welcome the submission of a revised version within three months for further consideration and would like to encourage you to address all the criticisms raised as suggested to improve conclusiveness and clarity. Please note that EMBO Molecular Medicine strongly supports a single round of revision and that, as acceptance or rejection of the manuscript will depend on another round of review, your responses should be as complete as possible.

I look forward to receiving your revised manuscript.

***** Reviewer's comments *****

Referee #1 (Comments on Novelty/Model System for Author):

I am in doubt whether this paper has sufficient impact, much of tumor promoting role of TGF-beta signaling in CAFs of pancreatic tumors was already known. Also found the paper really difficult to read, poor presentation and integration of the results.

Referee #1 (Remarks for Author):

In this paper the authors investigate the role of TGF-beta type 2 receptor (TbR2) in the stroma of pancreatic tumors. TbR2 is reported to mitigate IL6 secretion by CAFs, thereby inhibiting paracrine signaling of this cytokine on adjacent cancer cells reducing STAT3 activation. In addition, in PDAs in which cancer cells lack TbR2, targeting of TbR2 in stroma results in therapeutic benefit.

Major comments:

1. The authors need to be more clear in the text on the type of experiments performed, in which exact models and the reason for using that specific model. At present the reasoning and flow in the manuscript is difficult to follow. Many different murine and human models are used in a mixed way making it hard to interpret what the authors aim for and if the conclusion is in line with the presented experiment. KIC and KPC models have intact TGF-b signaling; why are not tumor models chosen with defective SMAD4 function that is common in human pancreatic cancer? NOD SCID and NSG mice are used that lack the adaptive immune system; does this not comprise the relevance of the obtained results with respect immune system?
2. Biffi et al 2018 (PMID: 30366930) needs to be discussed in the manuscript. It's surprising that this paper is not mentioned as it contains a very similar but also contradicting line of results concerning interleukin excretion by CAFs in overlapping models.
3. IL6 expression and excretion is affected by 2G8 challenge in the initial figures. However, the conclusion that TGFbeta promotes IL6 secretion by CAFs resulting in STAT3 activation in tumors drawn from figure 1D-F is based on circumstantial evidence and causality is not shown. There are more ways that STAT3 activity can be altered in tumors and the presence of IL6R only is no evidence for the proposed model. Additionally, this conclusion is opposite of the previously described negative effect of TGF on IL6 by biffi et al. In order to ensure that the authors proposed model is indeed correct, direct evidence needs to be collected.
4. scRNA-seq data used for figure panel 2 come from a non peer-reviewed Biorxiv manuscript. This makes it hard to assess the quality of the data and if the data is analyzed correctly. Therefore I suggest to either combine the dataset with this manuscript and let it be reviewed by an expert (which I am not). Or wait for the other manuscript to be accepted for publication.
5. The manuscript is focused on TbR2 expression/function. TbR2 functions in combination with TbRI; why the focus on TbR2, what about the expression of TbRI (in for example Fig. 2 analysis).
6. Page 7. It mentions: Activation of STAT3 by TGF-b-CM was sensitive too IL-6 blockade (Fig. 3E). I could not see the data regarding the IL-6 blockade.
7. In Figure 1C saline is used as control for 2G8 group. This is not a good control, a control, antibody of same isotype should have been used.
8. Figure 3. I do not understand the significance of this Figure; would it not be better to use the similar set up using conditioned media as used in Figure 3E? (and use phosphoSmad2 blot to check efficacy of TGFb treatment and cellular response).

Minor comments:

1. The authors incorrectly use the definition "organoids" to describe the cell line spheroid co-culture assays. The origin, complexity and culture conditions of (tumor)organoids are very different from the used procedure in the manuscript (PMID: 25035496).
2. There is no need to include Figure S5, total CDK4 blots are not a read-out for cell proliferation.
3. Include statistical analysis on data shown in Fig. 1b, supplementary Fig. 1, 2.
4. Include molecular weight markers on all Western blot results (Fig. 2C, 3A, 3H, 3I, 6A). in Fig. 2C the western blot is cut on two places. Provide explanation in the in Figure legend. Samples were analyzed on the same blot?
5. Some of the Western blot results are overexposed making the relative differences in expression difficult to analyze. Weaker exposures need to be shown for Fig. 2C, actin blot, 3I, actin/vimentin blot.
6. Figure 1. Show efficacy of 2G8 treatments with phosphoSMAD2 IHC.
7. Figure 2A, B. Colors that are used in the Figures need to be explained in the Figure legend.

Referee #2 (Remarks for Author):

This is an excellent study with high disease significance. Pancreatic is dismal disease in great need of new treatment approaches. The authors have provided a strong data set to support the translational value of targeting TGFbeta signaling in pancreatic cancer. The authors have used a combination of models and adequate number of controls to support the manuscript's major conclusions, however, additional experimentation is needed to define the specificity of the targeting and the mechanism controlling IL6 downstream of TGFbeta receptor.

- 1) It is important to determine that the effect of TGFbeta blockade is through the downregulation of IL6 signaling and its impact in tumor cell growth and immune response.
- 2) The immune profile should be further evaluated. There is a number of immune cell compartments that have not been investigated and are responsive to TGFbeta signaling and determine the level of immunosuppression in the tumors.
- 3) The mechanism of regulation of IL6 by TGFbeta should be evaluated. Is it mostly transcriptional? If so it will be important to demonstrate if Smad dependent or it is mostly through a non-canonical regulation, e.g., NFAT.

Referee #3 (Comments on Novelty/Model System for Author):

This is specifically addressed in detail in the review

Referee #3 (Remarks for Author):

Huang et al. have investigated the roles of stromal cell-associated TGFβ signaling in mediating the progression of PDA. The authors report that TGFβ-induced IL-6 secretion from cancer-associated fibroblasts stimulates tumor cell growth and inhibits NK activity via elevated STAT3 signaling. The authors go one to propose that stromal TGFβ signaling is a potential therapeutic target in PDA patients harboring inactivating TGFβ pathway mutations in the cancer cells.

The functions of stromal TGFβ in cancer and its clinical relevance are well known. The proposal to target stromal TGFβ signaling in cancer has been made many times before, based on different lines of evidence and different aspects of the biology of TGFβ in cancer, including TGFβ-induced production of IL-11 in CAFs to activate STAT3 signaling in cancer cells. Although the paper is not novel from this standpoint, the work would be suitable for publication provided that the mechanisms and cellular interactions involving TGFβ are well documented. The present manuscript falls a bit short in this regard, for the following reasons:

- 1) One major question of this manuscript is about how IL-6 derived from cancer-associated fibroblasts promotes tumor development. The authors proposed two mechanisms, one based on the activation of STAT3 in cancer cells to enhance their proliferation, and the other through the inhibition of NK cells by IL-6. However, the relative contribution of these two potential mechanisms is unclear and needs to be better defined. The authors should perform two key experiments: (1) compare the effect of 2G8 and neutralizing IL-6 antibody on tumor formation by wild-type and IL-6R knockout cancer cells in xenograft and syngeneic models, and (2) deplete NK cells in the NOD SCID model and determine the effect of IL-6 inhibition on tumor growth.
- 2) To further test the hypothesis of that IL-6 is the main TGF β -induced factor in CAFs which promotes tumor growth by activating STAT3 in cancer cells, the experiment in fig. 3f should include *Tgfr2* knockout cells or organoids, and comparing the effect of 2G8 and anti-IL-6 antibodies.
- 3) To test the hypothesis that TGF β -induced IL-6 promotes tumor growth by inhibiting NK cells, the authors compared tumor growth and metastasis in NOD SCID mice and NSG mice. However, the differences between these two mouse strains are not limited to NK cell levels. To determine the participation of NK cells the authors should compare NOD SCID mice with and without immunodepletion of NK cells. Also, the authors should compare tumor proliferation with or without 2G8 treatment to show how much the proposed TGF β /IL-6 axis promotes cancer cell growth without the complexity of immune interference. This is missing in fig 4.H-J.
- 4) In fig 3.e, it seems that TGF β treatment alone slightly inhibits and activates STAT3 phosphorylation in KIC and KPC, respectively. Could the authors comment on it?

1st Revision - authors' response

16 August 2019

Response to referee comments

A list of substantial changes made to the figures during revision:

1. Original Fig 1C IL-6 data from multiplex study in KIC has been moved to Fig S2. Mouse IL-6 ELISA was performed using KIC and KPC tumors treated with control antibody or 2G8. New data are shown in new Fig 1C-D.
2. In Fig 1E and H-I, P-SMAD2 staining in KIC and KPC tumors treated with control antibody or 2G8 has been added.
3. Original Fig 2A has been removed. scRNA-seq information on *Tgfr1* has been added to Fig 2A.
4. Original Fig 2C was repeated with additional cell lines (KPfC cells and pancreatic stellate cells) and now is Fig 2B.
5. IL-6 ELISA was performed in NIH3T3, pancreatic stellate cells, and two human CAF cell lines with TGF β and/or IL-1 treatment. Results are displayed as Fig 3D-F. Expression of different fibroblast markers in the two human CAF cell lines is shown in Fig S3B.
6. Fig 3G-H is new data on transcriptional control of IL-6 secretion by fibroblasts
7. Original Fig 3E has been repeated with an additional cell line derived from KPfC mouse and is now Fig 3J.
8. Original Fig 3F-H has been repeated with an additional cell line derived from KPfC mouse and is now Fig 3K-N.
9. Original Fig 3I has been moved to Fig S4.
10. Original Fig 4H-J has been moved to Fig S6. And the experiment was repeated using NK cell depleting antibody and shown in new Fig 4H-I. Validation of the NK cell depleting antibody is in Fig S5.
11. CD11b staining for MDSC and FOXP3 staining for regulatory T cells has been added to Fig 5.
12. Original Fig 5S-U is now Fig 6A-C.
13. Fig 8 is new and shows the contribution of TGF β -IL-6 paracrine signal on cancer cells and NK cells in the progression of TGF β R2 mutant PDA.
14. We have added a graphic synopsis as Fig S8.

Referee #1 (Comments on Novelty/Model System for Author):

I am in doubt whether this paper has sufficient impact, much of tumor promoting role of TGF-beta signaling in CAFs of pancreatic tumors was already known. Also found the paper really difficult to read, poor presentation and integration of the results.

Referee #1 (Remarks for Author):

In this paper the authors investigate the role of TGF-beta type 2 receptor (TbR2) in the stroma of pancreatic tumors. TbR2 is reported to mitigate IL6 secretion by CAFs, thereby inhibiting paracrine signaling of this cytokine on adjacent cancer cells reducing STAT3 activation. In addition, in PDAs in which cancer cells lack TbR2, targeting of TbR2 in stroma results in therapeutic benefit.

Major comments:

1. The authors need to be more clear in the text on the type of experiments performed, in which exact models and the reason for using that specific model. At present the reasoning and flow in the manuscript is difficult to follow. Many different murine and human models are used in a mixed way making it hard to interpret what the authors aim for and if the conclusion is in line with the presented experiment. KIC and KPC models have intact TGF-b signaling; why are not tumor models chosen with defective SMAD4 function that is common in human pancreatic cancer? NOD SCID and NSG mice are used that lack the adaptive immune system; does this not comprise the relevance of the obtained results with respect immune system?

Response: Thank you for your suggestion. We have modified our text and provided rationale for each model used in the studies.

2. Biffi et al 2018 (PMID: 30366930) needs to be discussed in the manuscript. It's surprising that this paper is not mentioned as it contains a very similar but also contradicting line of results concerning interleukin excretion by CAFs in overlapping models.

Response: We have added experiments (Fig. 2D-H) as well as discussion that address our results in context of the cited study. We found that TGFβ induced IL-6 secretion consistently in different types of fibroblasts in a JUND-dependent manner.

3. IL6 expression and excretion is affected by 2G8 challenge in the initial figures. However, the conclusion that TGFbeta promotes IL6 secretion by CAFs resulting in STAT3 activation in tumors drawn from figure 1D-F is based on circumstantial evidence and causality is not shown. There are more ways that STAT3 activity can be altered in tumors and the presence of IL6R only is no evidence for the proposed model. Additionally, this conclusion is opposite of the previously described negative effect of TGF on IL6 by biffi et al. In order to ensure that the authors proposed model is indeed correct, direct evidence needs to be collected.

Response: The goal of figure 1 was to identify the paracrine signal in the stroma that was altered by the 2G8 treatment. In subsequent figures we validate that TGFβ directly stimulated IL-6 secretion from different fibroblasts and also compared with IL-1 treatment that has been reported previously by Biffi et al.

4. scRNA-seq data used for figure panel 2 come from a non peer-reviewed Biorxiv manuscript. This makes it hard to assess the quality of the data and if the data is analyzed correctly. Therefore I suggest to either combine the dataset with this manuscript and let it be reviewed by an expert (which I am not). Or wait for the other manuscript to be accepted for publication.

Response: The scRNA-seq study has been published in JCI Insight. 2019 Jul 23;5. pii: 129212. doi: 10.1172/jci.insight.129212.

5. The manuscript is focused on TbR2 expression/function. TbR2 functions in combination with TbRI; why the focus on TbR2, what about the expression of TbRI (in for example Fig. 2 analysis).

Response: Thank you for the suggestion, we have shown TGFβR1 expression levels in Fig. 2. As you are aware, TGFβR2 is essential for TGFβ-mediated signaling via the heterodimeric complex of

TGF β R1/TGF β R2, thus TGF β R2 is a relevant target for interrupting TGF β signaling.

6. Page 7. It mentions: Activation of STAT3 by TGF- β -CM was sensitive too IL-6 blockade (Fig. 3E). I could not see the data regarding the IL-6 blockade.

Response: IL-6 blockade is shown in Fig. 3J (TGF β -CM+ α -IL-6) which reduced the STAT3 activation in primary mouse PDA cell lines.

7. In Figure 1C saline is used as control for 2G8 group. This is not a good control, a control, antibody of same isotype should have been used.

Response: In the original Fig. 1C, the control group was control rat IgG (isotype-matched) instead of saline. But we did use saline for Fig. 1A in which we performed the screening for stromal factors altered by 2G8. We agreed this was not the best control. However, we validated this initial screen with tumors treated with control rat IgG or 2G8 at the protein level and found the results were consistent (Fig. 1C-D).

8. Figure 3. I do not understand the significance of this Figure; would it not be better to use the similar set up using conditioned media as used in Figure 3E? (and use phosphoSmad2 blot to check efficacy of TGF β treatment and cellular response).

Response: It is not clear which figure the referee is referring to. However, briefly Fig. 3 demonstrates the connection between TGF β induction of IL-6 by fibroblasts and IL-6 stimulation of tumor cells. This signaling axis is evaluated at the level of protein expression, signaling and 3D co-culture growth.

Minor comments:

1. The authors incorrectly use the definition "organoids" to describe the cell line spheroid co-culture assays. The origin, complexity and culture conditions of (tumor)organoids are very different from the used procedure in the manuscript (PMID: 25035496).

Response: Thank you for pointing this out. We have altered the term using 3D culture/co-culture instead.

2. There is no need to include Figure S5, total CDK4 blots are not a read-out for cell proliferation.

Response: Thank you for the suggestion. We have removed the figure.

3. Include statistical analysis on data shown in Fig. 1b, supplementary Fig. 1, 2.

Response: These studies were ELISA arrays with limited number of samples. The goal was to screen for the factors in the stroma that were altered by 2G8 treatment. We further validated these results by IL-6 ELISA with higher sample numbers and performed statistical analysis on these results (P<0.0001).

4. Include molecular weight markers on all Western blot results (Fig. 2C, 3A, 3H, 3I, 6A). in Fig. 2C the western blot is cut on two places. Provide explanation in the in Figure legend. Samples were analyzed on the same blot?

Response: We have added molecular weight markers in all the western blots in this manuscript. In the original Fig.2B, different cell types were run separately. We have re-run the samples together on the same gel and also added additional types of cancer cells and fibroblasts (Fig. 2B).

5. Some of the Western blot results are overexposed making the relative differences in expression difficult to analyze. Weaker exposures need to be shown for Fig. 2C, actin blot, 3I, actin/vimentin blot.

Response: We have re-run blots for Fig. 2C with shorter exposure time. Original Fig. 3I has been moved to Fig. S4 and is shown with a lower exposure.

6. Figure 1. Show efficacy of 2G8 treatments with phosphoSMAD2 IHC.

Response: We have added these data. (Fig. 1E, H-I)

7. Figure 2A, B. Colors that are used in the Figures need to be explained in the Figure legend.

Response: We have removed Fig. 2A. There is no specific meaning of the colors used in the violin plots. Also, we have referred our scRNA-seq study for more details (JCI Insight. 2019 Jul 23;5. pii: 129212.).

Referee #2 (Remarks for Author):

This is an excellent study with high disease significance. Pancreatic is a dismal disease in great need of new treatment approaches. The authors have provided a strong data set to support the translational value of targeting TGFβ signaling in pancreatic cancer. The authors have used a combination of models and adequate number of controls to support the manuscript's major conclusions, however, additional experimentation is needed to define the specificity of the targeting and the mechanism controlling IL6 downstream of TGFβ receptor.

Response: Thank you for your enthusiasm for the study.

1) It is important to determine that the effect of TGFβ blockade is through the downregulation of IL6 signaling and its impact in tumor cell growth and immune response.

Response: Thank you for the suggestion. We have addressed these questions with new Fig. 8, which demonstrates that IL-6 signaling is important on cancer cells and NK cells. This we believe significantly strengthens the impact of the study.

2) The immune profile should be further evaluated. There is a number of immune cell compartments that have not been investigated and are responsive to TGFβ signaling and determine the level of immunosuppression in the tumors.

Response: Thank you for the suggestion. We have added evaluation on CD11b+ MDSC and regulatory T cells in Fig. 5. Future studies will move towards incorporation of IO strategies and for those studies expanded immune landscape analysis will be performed.

3) The mechanism of regulation of IL6 by TGFβ should be evaluated. Is it mostly transcriptional? If so it will be important to demonstrate if Smad dependent or it is mostly through a non-canonical regulation, e.g., NFAT.

Response: This was a great suggestion. We investigated the transcription factors associated with the canonical and non-canonical TGFβ pathways and found the effect was mediated by the non-canonical transcription factor JUND (Fig. 3G). Again this significantly strengthens the study, we appreciate the insight.

Referee #3 (Comments on Novelty/Model System for Author):

This is specifically addressed in detail in the review

Referee #3 (Remarks for Author):

Huang et al. have investigated the roles of stromal cell-associated TGFβ signaling in mediating the progression of PDA. The authors report that TGFβ-induced IL-6 secretion from cancer-associated fibroblasts stimulates tumor cell growth and inhibits NK activity via elevated STAT3 signaling. The authors go on to propose that stromal TGFβ signaling is a potential therapeutic target in PDA patients harboring inactivating TGFβ pathway mutations in the cancer cells.

The functions of stromal TGF β in cancer and its clinical relevance are well known. The proposal to target stromal TGF β signaling in cancer has been made many times before, based on different lines of evidence and different aspects of the biology of TGF β in cancer, including TGF β -induced production of IL-11 in CAFs to activate STAT3 signaling in cancer cells. Although the paper is not novel from this standpoint, the work would be suitable for publication provided that the mechanisms and cellular interactions involving TGF β are well documented. The present manuscript falls a bit short in this regard, for the following reasons:

1) One major question of this manuscript is about how IL-6 derived from cancer-associated fibroblasts promotes tumor development. The authors proposed two mechanisms, one based on the activation of STAT3 in cancer cells to enhance their proliferation, and the other through the inhibition of NK cells by IL-6. However, the relative contribution of these two potential mechanisms is unclear and needs to be better defined. The authors should perform two key experiments: (1) compare the effect of 2G8 and neutralizing IL-6 antibody on tumor formation by wild-type and IL-6R knockout cancer cells in xenograft and syngeneic models, and (2) deplete NK cells in the NOD SCID model and determine the effect of IL-6 inhibition on tumor growth.

Response: This was a great suggestion. We have investigated this with IL6R knockdown and NK cell depletion (Fig. 8). Figure 8 demonstrates IL-6 signaling on cancer cells and NK cells contributes to tumor development and progression. In our opinion these data bolster the impact of the data.

2) To further test the hypothesis of that IL-6 is the main TGF β -induced factor in CAFs which promotes tumor growth by activating STAT3 in cancer cells, the experiment fig. 3f should include Tgfr2 knockout cells or organoids, and comparing the effect of 2G8 and anti-IL-6 antibodies.

Response: Thank you for the suggestion. We have explored this question in Fig. 8B, which suggests TGF β -induced IL-6 directly stimulates cancer cells to promote tumor growth.

3) To test the hypothesis that TGF β -induced IL-6 promotes tumor growth by inhibiting NK cells, the authors compared tumor growth and metastasis in NOD SCID mice and NSG mice. However, the differences between these two mouse strains are not limited to NK cell levels. To determine the participation of NK cells the authors should compare NOD SCID mice with and without immunodepletion of NK cells. Also, the authors should compare tumor proliferation with or without 2G8 treatment to show how much the proposed TGF β /IL-6 axis promotes cancer cell growth without the complexity of immune interference. This is missing in fig4.H-J.

Response: Thank you for the suggestion. We have repeated the experiment using NK cell depletion and found similar results. IL-6 is known to be a species-specific cytokine (Proteins. 1997 Jan;27(1):96-109.), therefore, in the human xenograft models, IL-6 from mouse cells did not affect the human cancer cells directly. However, we evaluated the direct effect of IL-6 on cancer cells in Fig. 8 using 3D co-culture system and also by expression of Ki67 in vivo.

4) In fig3.e, it seems that TGF β treatment alone slightly inhibits and activates STAT3 phosphorylation in KIC and KPC, respectively. Could the authors comment on it?

Response: We repeated the experiments with a PDA cell line derived from a *KP/C* mouse, which is P53-deficient. We notice that TGF β slightly activated STAT3 in cell lines that have lost P53 function (*KPC* and *KP/C*). Future studies are called for to understand this phenomenon. Regarding the effect of TGF β on STAT3 in *KIC* cells, the result of slight inhibition of STAT3 by 2G8 is consistent in vitro but the molecular mechanism of this slight effect is unclear. In vivo we find that 2G8 reduces STAT3 in tumor cells regardless of genotype of the tumor cells (Fig. 1F).

2nd Editorial Decision

6 September 2019

Thank you for the submission of your revised manuscript to EMBO Molecular Medicine. We have now received the enclosed reports from the referees that were asked to re-assess it. As you will see the reviewers are now supportive and I am pleased to inform you that we will be able to accept your manuscript pending minor editorial amendments.

I look forward to reading a new revised version of your manuscript within two weeks.

***** Reviewer's comments *****

Referee #1 (Remarks for Author):

The authors have improved the manuscript. In particular biological and clinical relevance have become stronger in the revised version. Molecular mechanistically it could have been more developed. However, also considering the comments by other two reviewers, I think the paper is now acceptable for publication.

Referee #2 (Comments on Novelty/Model System for Author):

None

Referee #2 (Remarks for Author):

No further comments.

Referee #3 (Remarks for Author):

My concerns have been addressed by the authors. The current manuscript has been greatly improved and I recommend it for publication

Corresponding Author Name: Roif A. Brekken
 Journal Submitted to: EMBO Molecular Medicine
 Manuscript Number: EMM-2019-10515